

# Chemistry and deposition in the Model of Atmospheric composition at Global and Regional scales using Inversion Techniques for Trace gas Emissions (MAGRITTE v1.0). Part B. Dry deposition

Jean-François Müller[1], Trissevgeni Stavrakou[1], Maite Bauwens[1], Steven Compernolle[1], and Jozef Peeters[2]

[1]Royal Belgian Institute for Space Aeronomy, Avenue Circulaire 3, 1180, Brussels, Belgium
[2]Department of Chemistry, University of Leuven, Celestijnenlaan 200F, B-3001 Leuven, Belgium

**Correspondence:** Jean-François Müller (Jean-Francois.Muller@aeronomie.be)

**Abstract.** A new module for calculating the dry deposition of trace gases is presented and implemented in the Model of Atmospheric composition at Global and Regional scales using Inversion Techniques for Trace gas Emissions (MAGRITTE v1.0). The dry deposition velocities are calculated using Wesely's classical resistance-in-series approach. While relying on analyses of the European Centre for Medium-range Weather Forecasts (ECMWF) for meteorological fields, the aerodynamic resistance calculation module is based on the ECMWF model equations for turbulent transfer within the surface layer. The stomatal resistance for water vapour is calculated using a Jarvis-type parameterization in a multi-layer canopy environment model accounting for the leaf area index (LAI). The gas-phase diffusion coefficients needed to relate the stomatal resistances of different species are calculated from molecular structure. The cuticular, mesophyll and soil resistances depend on the species reactivity and Henry's Law constant (HLC). The HLCs of organic species for which no experimental data is available are estimated using a newly-developed prediction method based on existing methods for vapour pressures (EVAPORATION, Estimation of VApour Pressure of Organics) and infinite dilution activity coefficients (AIOMFAC, Aerosol Inorganic Organic Mixtures Functional groups Activity Coefficients).

Acknowledging the dominance of stomatal uptake for ozone dry deposition, the stomatal resistance model parameters for 6 of the 7 major plant functional types (PFT) are adjusted based on extensive model comparisons with field measurements of ozone deposition velocity at 24 sites worldwide. The modelled OVOC deposition velocities for 25 different OVOCs are evaluated against field data from a total of 20 studies. The comparison shows the need for a species-dependent adjustment of the canopy resistances in order to match the observed variability among different species. This is realized by multiplying the HLC of each OVOC by a species-dependent parameter $f_1$ adjusted based on the comparisons. The values of $f_1$ span a wide range, from values of the order of unity or less for formaldehyde and several trifunctional compounds, to $>10^4$ for compounds seen to deposit rapidly despite their low water-solubility, like MVK, MACR, $CH_3CHO$ and PAN. Despite the acknowledged caveats of the approach, the resulting modelled deposition velocities are consistent with the existing experimental data. The results of global-scale MAGRITTE model simulations demonstrate the importance of OVOC dry deposition on their global





abundance. It is found to remove from the atmosphere the equivalent of 27% of the global NMVOC emissions on a carbon basis, as well as about 8% of NOx emissions in the form of organic nitrates and PAN-like compounds.

## 1 Introduction

Dry deposition is a major removal process for a large number of atmospheric pollutants such as ozone, hydrogen perox-
ide, and sulfur and nitrogen oxides (Wesely, 1989). Being also an important sink of the oxygenated volatile organic compounds (OVOCs) (Karl et al., 2010), it affects the abundance of organic and inorganic acidifying substances (Brook et al., 1999; Sanhueza et al., 1996), the fluxes of organic and inorganic nitrogen to ecosystems (Paulot et al., 2018), the reactivity of the hydroxyl radical (Nölscher et al., 2016) and the formation of secondary organic aerosol (SOA) (Glasius and Goldtsein, 2016). Whereas the dry deposition of submicron particles such as SOA is very slow (Seinfeld and Pandis, 2006), the deposi-
tion of gas-phase semi-volatile SOA precursors is believed to have a very large impact on SOA levels, especially in the case of biogenically-derived SOA owing to their expected low vapour pressures and high solubility in water (Hodzic et al., 2014; Knote et al., 2015; Shrivasta et al., 2015).

The deposition velocity of a compound (traditionally given in cm s$^{-1}$), i.e. the ratio of its downward mass flux to its concentration, is a complex function of meteorology (in particular turbulence), chemical compound properties, and surface type and
properties. It is usually modelled using the resistance-in-series approach Wesely (1989), decomposing the overall resistance to deposition into contributions from atmospheric turbulence (aerodynamic resistance), molecular diffusion (quasi-laminar boundary layer resistance) and surface processes (surface resistance). The surface resistance to deposition of a gaseous compound depends on its reactivity and water-solubility. Due to the scarcity of field data, in particular for OVOCs, the surface resistance for any compound was expressed by Wesely as a combination of parameterized resistances for two model com-
pounds, O$_3$ (template for very reactive species) and SO$_2$ (template for soluble species), for which abundant dry deposition field data is available. Due to the much lower reactivity of OVOCs in comparison with ozone, their predicted deposition velocities were generally very low, to the exception of very soluble compounds like formic acid.

The observation of fast deposition of several OVOCs to both temperate and tropical forest ecosystems challenged that view (Karl et al., 2010), leading to the recommendation that most OVOCs should be considered as being as reactive as ozone when
calculating their surface resistance. In particular, the OVOCs were suggested to be lost immediately upon entering the stomata, just like ozone. This simple recipe for evaluating OVOC deposition velocities has been in use in many models (e.g. Fischer et al. (2014); Wells et al. (2014)). However, the measurement of dry deposition velocities for a large suite of OVOCs at a deciduous forest in Alabama (Nguyen et al., 2015) led to a different conclusion seemingly at odds with the reactivity-driven deposition model of Karl et al. (2010): the most water-soluble compounds deposit much faster than moderately soluble ones, and the least
soluble species showed little uptake. This result implies that the Henry's law constant should play a more important role in the resistance calculation than previously considered in Wesely's model or its adaptation by Karl et al.. Although Nguyen et al. (2015) proposed a model adaptation providing a satisfactory agreement with their observations, it was not reconciled with previous field measurements for OVOCs and other compounds.





Here we present a new dry deposition scheme based on Wesely's approach, incorporating an updated representation of the different resistances. The model is evaluated and to a large degree constrained by available field data. In particular, the dependence of the canopy conductance on the Henry's law constant (HLC) is modified through the introduction of a unit-less, species-dependent factor $f_1$ multiplying the HLC in the expressions of the mesophyll, ground and cuticle resistances.

Acknowledging the importance of HLC estimation for the resistance parameterization, a new HLC prediction method is presented and evaluated against laboratory measurements and previous estimation methods (Sect. 2 and the Supplement, see http://tropo.aeronomie.be/index.php/models/magritte). Next, the modules for the calculation of aerodynamic, quasi-laminar boundary layer and surface resistances are described in detail (Sect. 3.1-3.3), and their evaluation against field data for ozone, sulfur dioxide and the OVOCs is presented in Sect. 3.4 and 3.5. The new scheme is implemented in the Model of Atmospheric

composition at Global and Regional scales using Inversion Techniques for Trace gas Emissions (MAGRITTE), of which the biogenic volatile organic compound (BVOC) oxidation mechanism is described in a companion paper (Müller et al., 2018). Model simulations at global and regional scales are conducted to provide an updated evaluation of the role of dry deposition in the budget of OVOCs and as a sink of NOx. Finally, Sect. 5 recapitulate the main findings and provide a tentative summary of the uncertainties and limitations of the current approach.

**2 Henry's law constants of organic model compounds**

Both the wet deposition and dry deposition parameterizations rely on the estimation of Henry's law constant of the gaseous model compounds. The wet scavenging scheme in MAGRITTE is based on cloud and precipitation fields provided by the European Centre for Medium-range Weather Forecasts (ECMWF), and is described in detail in Stavrakou et al. (2009b). It distinguishes washout by convective precipitation (included in the convective transport scheme) from scavenging in and below

large-scale stratiform clouds, which is represented as a first-order process. Henry's law is used to calculate the gas-aqueous partitioning of gaseous compounds in clouds containing liquid water. Note that the model now uses an increased scavenging efficiency (0.52) for methyl hydroperoxide in convective updrafts based on a recent field data analysis (Barth et al., 2016). This value is also used as lower limit for the scavenging efficiency of other hydroperoxides. Furthermore, peroxides are assumed to be entirely retained in frozen cloud particles during freezing (Barth et al., 2016).

For many compounds, experimental HLC estimates exist (Sander, 2015); however, for most compounds of the chemical mechanism (Table 1), estimation methods are required. There are several ways of describing the solubility in water. Here, the HLC ($K_H$ in mol L$^{-1}$ atm$^{-1}$) is defined as the equilibrium ratio of the aqueous phase concentration ($c_a$ in mol L$^{-1}$) to the partial pressure ($p_g$ in atm) of the considered compound,

$$K_H = \frac{c_a}{p_g}. \tag{1}$$

The HLC can be estimated using

$$K_H = \frac{c_w}{p_L^0 \, \gamma^\infty} \tag{2}$$





with $p_L^0$ the liquid state saturation vapour pressure (atm), $c_w$ the concentration of pure water (55.5 mol L$^{-1}$ at room temperature) and $\gamma^\infty$ the infinite dilution activity coefficient (IDAC) of the compound in water. $K_H$ can be estimated directly, or by separate estimations of $p_L^0$ and $\gamma^\infty$. Several HLC estimation methods are evaluated in this work, as detailed in the Supplement (http://tropo.aeronomie.be/index.php/models/magritte). The method retained for use in MAGRITTE relies on Eq. (2), with

vapour pressure ($p_L^0$) estimates obtained from the group-contribution method EVAPORATION (Estimation of VApour Pressure of Organics) (Compernolle et al., 2011), and infinite dilution activity coefficient estimates based on AIOMFAC (Aerosol Inorganic Organic Mixtures Functional groups Activity Coefficients) (Zuend et al., 2011), with several modifications.

EVAPORATION and other vapour pressure estimation methods were evaluated by O'Meara et al. (2014). Within its scope, EVAPORATION was found to provide the most accurate $p_L^0$ estimations. AIOMFAC can be considered as a generalisation

of UNIFAC (UNIQUAC Functional-group Activity Coefficients) (Fredenslund et al., 1977). Regarding the organic part, it combines features of the UNIFAC versions of Peng et al. (2001) and Marcolli et al. (2005), both developed specifically for atmospherically relevant compounds. AIOMFAC, in combination with a vapour pressure method, has the widest scope of all methods considered here. We use here a modified version of AIOMFAC, denoted AIOMFAC(m), incorporating modifications to the acid, nitrate and peroxynitrate interaction parameters (see Supplement). Furthermore, since AIOMFAC does not take into

account the effect of hydration explictly for carbonyls, we replace $\gamma^\infty$ in Eq. (2) by $\gamma^\infty/F_{\text{hyd}}$, $F_{\text{hyd}}$ being the ratio of effective over intrinsic HLC as calculated by the method of Raventos-Duran et al. (2010). For molecules with no carbonyl functionality, this factor is equal to unity. However, for glyoxal, $F_{\text{hyd}}$ reaches $6 \times 10^4$. Note that this procedure slightly degrades the results for mono-aldehydes, likely because hydration was implicitly taken into account in the interaction parameters for those compounds.

The above approach has its limitations. Most importantly, all group-contribution methods are believed to generally underpre-

dict the saturation vapour pressures of highly oxidized compounds, due to the still limited scope of their basis sets of empirical data and the lack of interaction parameters accounting for the effect of hydrogen bonding (Valorso et al., 2011; Kurtén et al., 2016).

The HLC values used in the model are provided in Table 1. When available, experimental data are used, obtained from the review of Sander (2015). Otherwise, the values are calculated as described above. To account for acid dissociation, the HLC of

carboxylic acids in Table 1 is multiplied by the factor $(1 + K_a/[\text{H}^+])$ with dissociation constants ($K_a$) of $1.8 \times 10^{-4}$, $1.7 \times 10^{-5}$ and $1.3 \times 10^{-5}$ mol L$^{-1}$ for HCOOH, CH$_3$COOH, and C$_2$H$_5$COOH, respectively (Lide, 2000). Cloud water pH, relevant to the wet scavenging rate estimation, is taken equal to 4.5; the pH of water in oceans and lakes, relevant to the dry deposition scheme, is assumed to be near-neutral (pH = 7) (Wesely, 1989). The effect of acid dissociation is not taken into account in the parameterization of dry deposition over land, as detailed in Sect. 3.5.6.





**Table 1.** Henry's law constant (HLC, $K_H = K_{H,298} \exp(B/T - B/298)$, in mol $L^{-1}$ atm$^{-1}$) of OVOCs in MAGRITTE (including effect of carbonyl hydration, but not acid dissociation), molecular weight (MW), gas-phase diffusivity at 298 K and 1 atm ($D_{g,r}$, in cm$^2$ s$^{-1}$) and values of $f_0$ and $f_1$ parameters of the surface resistance scheme (Sect. 3.3.4)). Also provided is the value of $\alpha = f_1 \cdot K_{H,298}/10^5$ at 298 K (see Sect. 3.3.4). Read 8.9(3) as $8.9 \cdot 10^3$. References for HLC: 1, Sander (2015); 2, this work (see text); Note: a, assume $K_H = K_H(CH_3OOH)$.

| Compound | Chemical formula | $K_{H,298}$ | $B$ | MW | $D_{g,r}$ | $f_0$ | $f_1$ | $\alpha$ | Ref. |
|---|---|---|---|---|---|---|---|---|---|
| | *monofunctional compounds* | | | | | | | | |
| HCHO | HCHO | 3200 | 7100 | 30 | 0.172 | 0 | 0.03 | 0.001 | 1 |
| $CH_3CHO$ | $CH_3CHO$ | 13 | 5900 | 44 | 0.128 | 0 | 2(4) | 2.6 | 1 |
| $CH_3OH$ | $CH_3OH$ | 200 | 5600 | 32 | 0.166 | 0 | 600 | 1.2 | 1 |
| $C_2H_5OH$ | $C_2H_5OH$ | 190 | 6400 | 46 | 0.097 | 0 | 600 | 1.14 | 1 |
| $CH_3OOH$ | $CH_3OOH$ | 310 | 5200 | 48 | 0.138 | 0.1 | 400 | 1.24 | 1 |
| $CH_3OOOH$ | $CH_3OOOH$ | 310 [a] | 5200[a] | 64 | 0.124 | 0 | 400 | 1.24 | |
| VA | $CH_2{=}CHOH$ | 44 | 3800 | 44 | 0.128 | 0 | 600 | 0.26 | 2 |
| MVA | $CH_2{=}C(CH_3)OH$ | 44 | 3800 | 58 | 0.106 | 0 | 600 | 0.26 | 2 |
| $CH_3COCH_3$ | $CH_3COCH_3$ | 27 | 5500 | 58 | 0.106 | 0 | 2(3) | 0.54 | 1 |
| $CH_3ONO_2$ | $CH_3ONO_2$ | 2.0 | 4700 | 77 | 0.118 | 0 | 1(3) | 0.02 | 1 |
| PAN | $CH_3CO_3NO_2$ | 4.1 | 5700 | 121 | 0.093 | 1 | 1.5(4) | 0.62 | 1 |
| HCOOH | HCOOH | 8900 | 6100 | 46 | 0.153 | 0 | 20 | 1.78 | 1 |
| $CH_3COOH$ | $CH_3COOH$ | 4000 | 6200 | 60 | 0.124 | 0 | 20 | 0.8 | 1 |
| $C_2H_5COOH$ | $C_2H_5COOH$ | 4500 | 6800 | 74 | 0.095 | 0 | 20 | 0.9 | 1 |
| MCOOH | $CH_2{=}CH(CH_3)COOH$ | 5000 | 6795 | 87 | 0.088 | 1 | 20 | 1.0 | 2 |
| MVK | $CH_2{=}CHCOCH_3$ | 26 | 4800 | 70 | 0.074 | 0 | 5(4) | 13 | 1 |
| MACR | $CH_2{=}CCH_3CHO$ | 6.3 | 4600 | 70 | 0.074 | 0 | 5(4) | 3.2 | 1 |
| MPAN | $CH_2{=}CCH_3CO_3NO_2$ | 1.7 | 5700 | 147 | 0.077 | 1 | 5(4) | 0.85 | 1 |
| PAA | $CH_3COOOH$ | 840 | 5300 | 76 | 0.085 | 0.1 | 5(3) | 42 | 1 |
| MCO3H | $CH_2{=}CH(CH_3)CO(OOH)$ | 115 | 5257 | 103 | 0.107 | 0.1 | 5(3) | 5.8 | 2 |
| | *difunctional compounds* | | | | | | | | |
| GLYALD | $CH_2OHCHO$ | 4.1(4) | 4600 | 60 | 0.116 | 0 | 20 | 8.2 | 1 |
| HYAC | $CH_2OHCOCH_3$ | 7.8(3) | 6397 | 74 | 0.098 | 0 | 100 | 7.8 | 1,2 |
| HCOC5 | $CH_2{=}C(CH_3)C(O)CH_2OH$ | 3.0(3) | 6072 | 100 | 0.080 | 0 | 100 | 3.0 | 1 |
| HMHP | $HOCH_2OOH$ | 1.7(6) | 9900 | 64 | 0.124 | 0.1 | 50 | 1700 | 1 |
| ISOPBOOH | $CH_2{=}CHC(CH_3)(OOH)CH_2OH$ | 5.0(4) | 8170 | 118 | 0.076 | 0.1 | 50 | 25 | 2 |
| ISOPDOOH | $CH_2{=}C(CH_3)CH(OOH)CH_2OH$ | 7.4(4) | 8381 | 118 | 0.076 | 0.1 | 50 | 37 | 2 |
| ISOPEOOH | $CH_2{=}C(CH_3)CH(OH)CH_2OOH$ | 2.4(4) | 7617 | 118 | 0.076 | 0.1 | 50 | 12 | 2 |
| MVKOOH | $0.55\,CH_3C(O)CH(OOH)CH_2OH$ | 4.6(6) | 9652 | 120 | 0.080 | 0.1 | 50 | 2300 | 2 |

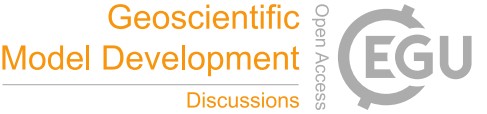



| Compound | Chemical formula | $K_{H,298}$ | $B$ | MW | $D_{g,r}$ | $f_0$ | $f_1$ | $\alpha$ | Ref. |
|---|---|---|---|---|---|---|---|---|---|
| | $+0.45\,CH_3C(O)CH(OH)CH_2OOH$ | | | | | | | | |
| NISOPOOHD | $0.84\,HOOCH_2CH{=}C(CH_3)CH_2ONO_2$ | 2.9(3) | 8372 | 163 | 0.070 | 0.1 | 400 | 12 | 2 |
| | $+0.16\,O_2NOCH_2CH{=}C(CH_3)CH_2OOH$ | | | | | | | | |
| NISOPOOHB | $0.9\,CH_2{=}CHC(CH_3)(OOH)CH_2ONO_2$ | 7.9(2) | 7700 | 163 | 0.070 | 0.1 | 400 | 3.2 | 2 |
| | $+0.1\,CH_2{=}C(CH_3)CH(OOH)CH_2ONO_2$ | | | | | | | | |
| GLY | $CHOCHO$ | 4.2(5) | 7500 | 58 | 0.120 | 0 | 20 | 84 | 1 |
| MGLY | $CH_3COCHO$ | 2.4(4) | 7100 | 72 | 0.101 | 0 | 20 | 4.8 | 1 |
| PYRA | $CH_3COCOOH$ | 3.1(5) | 5100 | 88 | 0.097 | 0 | 30 | 93 | 1 |
| GCOOH | $HOCH_2COOH$ | 1.8(7) | 10042 | 76 | 0.109 | 0 | 30 | 5400 | 2 |
| GCO3H | $HOCH_2CO(OOH)$ | 4.8(5) | 8377 | 92 | 0.100 | 0.1 | 5(3) | 24000 | 2 |
| ISOPBNO3 | $CH_2{=}CHC(CH_3)(ONO_2)CH_2OH$ | 1.1(3) | 7138 | 147 | 0.072 | 0 | 1(3) | 11 | 2 |
| ISOPDNO3 | $CH_2{=}C(CH_3)CH(ONO_2)CH_2OH$ | 1.3(3) | 7325 | 147 | 0.072 | 0 | 1(3) | 13 | 2 |
| ISOPCNO3 | $0.86\,HOCH_2CH{=}C(CH_3)CH_2ONO_2$ | 3.1(3) | 7797 | 147 | 0.072 | 0 | 1(3) | 31 | 2 |
| | $+0.14\,HOCH_2C(CH_3){=}CHCH_2ONO$ | | | | | | | | |
| ISOPENO3 | $CH_3C({=}CH_2)CH(OH)CH_2ONO_2$ | 2.8(2) | 6512 | 147 | 0.072 | 0 | 1(3) | 2.8 | 2 |
| APINONO2 | $C_{10}H_{16}(OH)(ONO_2)$ | 3.2(2) | 9079 | 231 | 0.056 | 0 | 1(3) | 3.2 | 2 |
| NOA | $CH_3C(O)CH_2ONO_2$ | 1.0(3) | 5312 | 119 | 0.087 | 0 | 1(3) | 10 | 1,2 |
| ISOPBOH | $CH_2{=}CHC(CH_3)(OH)CH_2OH$ | 1.5(5) | 8029 | 102 | 0.079 | 0 | 600 | 900 | 2 |
| ISOPDOH | $CH_2{=}C(CH_3)CH(OH)CH_2OH$ | 6.3(4) | 7732 | 102 | 0.079 | 0 | 600 | 380 | 2 |
| NC4CHO | $0.75\,OCHCH{=}C(CH_3)CH_2ONO_2$ | 2.0(3) | 6330 | 145 | 0.073 | 0 | 1(3) | 20 | 2 |
| | $+0.25\,OCHC(CH_3){=}CHCH_2ONO_2$ | | | | | | | | |
| GPAN | $HOCH_2CO_3NO_2$ | 7.9(3) | 7179 | 137 | 0.089 | 1 | 1.5(4) | 1200 | 2 |
| MVKNO3 | $0.8\,CH_3COCH(ONO_2)CH_2OH$ | 1.8(5) | 9080 | 149 | 0.076 | 0 | 6 | 11 | 2 |
| | $+0.2\,CH_3COCH(OH)CH_2ONO_2$ | | | | | | | | |
| MACRNO3 | $OCHC(CH_3)(ONO_2)CH_2OH$ | 1.2(7) | 8720 | 149 | 0.076 | 0 | 6 | 720 | 2 |
| ETHLN | $OCHCH_2ONO_2$ | 3.1(3) | 4644 | 105 | 0.098 | 0 | 1(3) | 31 | 2 |
| HPALD1 | $OCHC(CH_3){=}CHCH_2(OOH)$ | 2.7(4) | 7343 | 116 | 0.077 | 0.1 | 30 | 8.1 | 2 |
| HPALD2 | $OCHCH{=}C(CH_3)CH_2(OOH)$ | 2.0(5) | 8835 | 116 | 0.077 | 0.1 | 30 | 60 | 2 |
| HPACET | $CH_3COCH_2OOH$ | 3.1(3) | 6178 | 90 | 0.093 | 0.1 | 30 | 0.93 | 2 |
| HPAC | $OCHCH_2OOH$ | 7.6(3) | 5469 | 76 | 0.107 | 0.1 | 30 | 2.3 | 2 |
| HMAC | $OCHC(CH_3){=}CHOH$ | 1.9(4) | 6445 | 86 | 0.089 | 0 | 100 | 19 | 2 |
| HMVK | $CH_3C(O)CH{=}CHOH$ | 1.3(4) | 6888 | 86 | 0.089 | 0 | 100 | 13 | 2 |
| | *polyfunctional compounds* | | | | | | | | |
| IEPOX | $HOCH_2\overline{CHOC}(CH_3)CH_2OH$ | 4.5(6) | 10484 | 118 | 0.076 | 0 | 50 | 2250 | 2 |



| Compound | Chemical formula | $K_{H,298}$ | $B$ | MW | $D_{g,r}$ | $f_0$ | $f_1$ | $\alpha$ | Ref. |
|---|---|---|---|---|---|---|---|---|---|
| ICHE | $HOCH_2CHOC(CH_3)CHO$ | 5.0(6) | 8100 | 116 | 0.077 | 0 | 50 | 2500 | 2 |
| IHNE | $0.57\ O_2NOCH_2C(CH_3)OCHCH_2OH$ | 4.6(3) | 9875 | 163 | 0.070 | 0 | 50 | 2.3 | 2 |
|  | $+0.43\ O_2NOCH_2C(CH_3)(OH)CHOCH_2$ |  |  |  |  |  |  |  |  |
| HMML | $HOCH_2C(CH_3)OC=O$ | 8.5(3) | 7820 | 102 | 0.085 | 0 | 100 | 8.5 | 2 |
| MMAL | $O=CCH=C(CH_3)C(=O)O$ | 2.2(3) | 6880 | 112 | 0.088 | 0 | 100 | 2.2 | 2 |
| DHA | $CH_3C(O)CH(OH)_2$ | 1.3(5) | 7286 | 90 | 0.093 | 0 | 1 | 1.3 | 2 |
| DHBO | $CH_3C(O)CH(OH)CH_2OH$ | 3.3(6) | 9291 | 104 | 0.083 | 0 | 0.1 | 3.3 | 2 |
| MACROH | $HOCH_2C(CH_3)(OH)CHO$ | 2.4(6) | 7677 | 104 | 0.083 | 0 | 0.1 | 2.4 | 2 |
| HOBA | $CH_3C(O)CH(OH)CHO$ | 1.1(6) | 6670 | 102 | 0.085 | 0 | 1 | 11 | 2 |
| BIACETOH | $CH_3COCOCH_2OH$ | 2.3(5) | 7790 | 102 | 0.085 | 0 | 1 | 2.3 | 2 |
| INCCO | $HOCH_2C(O)C(CH_3)(OH)CH_2ONO_2$ | 4.2(6) | 10051 | 179 | 0.068 | 0 | 1 | 42 | 2 |
| INCNO3 | $HOCH_2CH(ONO_2)C(CH_3)(OH)CH_2ONO_2$ | 1.1(7) | 12374 | 226 | 0.064 | 0 | 1 | 110 | 2 |
| MBONO3 | $0.67\ CH_3C(OH)(CH_3)CH(ONO_2)CH_2OH$ | 3.0(5) | 9533 | 165 | 0.069 | 0 | 1 | 3.0 | 2 |
|  | $+0.33\ CH_3C(OH)(CH_3)CH(OH)CH_2ONO_2$ |  |  |  |  |  |  |  |  |
| INDOOH | $HOCH_2CH(ONO_2)C(CH_3)(OOH)CH_2OH$ | 1.4(10) | 15052 | 197 | 0.066 | 0.1 | 1 | 1.4(5) | 2 |
| DIHPCHO | $CH_3C(OOH)(CHO)CH_2OOH$ | 2.2(7) | 9459 | 136 | 0.077 | 0.1 | 1 | 220 | 2 |
| DIHPMEK | $CH_3C(O)CH(OOH)CH_2OOH$ | 5.0(6) | 9975 | 136 | 0.077 | 0.1 | 1 | 50 | 2 |
| HPKETAL | $CH_3C(O)CH(OOH)CHO$ | 1.8(6) | 7318 | 118 | 0.082 | 0.1 | 1 | 18 | 2 |
| HPDIAL | $OCHC(CH_3)(OOH)CHO$ | 4.9(7) | 6969 | 118 | 0.082 | 0.1 | 1 | 490 | 2 |



## 3  Dry deposition scheme

Following e.g. Wesely (1989), the dry deposition velocity ($V_d$) is expressed as

$$V_d = \frac{1}{R_a + R_b + R_c} \tag{3}$$

with $R_a$ the aerodynamic resistance between the surface and a specified height (taken here to be the elevation of the first model level,
approximately 10 m), $R_b$ the quasi-laminar sublayer resistance (controlled by the diffusivity of the gaseous compound in air), and $R_c$ the bulk
surface resistance, dependent on properties of both the surface and the chemical compound. Whereas $R_a$ is identical for all species, $R_b$ and
$R_c$ are species-dependent. The resistances $R_a$ and $R_b$ depend on meteorological quantitites (Monin-Obukhov length, friction velocity, etc.)
which are calculated using parameterizations of the ECMWF Integrated Forecasting System (IFS) (ECMWF, 2014), as detailed hereafter.

### 3.1  Aerodynamic resistance, $R_a$

The vertical fluxes of chemical compounds in the lowermost part of the planetary boundary layer are well represented by formulations based
on the Monin-Obukhov (M-O) similarity theory, which describes relationships between vertical profiles and fluxes using a metric called the
Obukhov length ($L$) (ECMWF, 2014; Toyota et al., 2016)

$$L = -\frac{u_*^3\, T_l}{\kappa\, g\, Q_{0\nu}} \tag{4}$$

with $u_*$ is the friction velocity (m s$^{-1}$), $T_l$ the temperature at the lowermost level (K), $\kappa$ von Karman's constant (=0.41), $g$ the acceleration
due to gravity (m s$^2$), and $Q_{0\nu}$ the virtual temperature flux in the surface layer, related to the sensible heat flux $S$ (W m$^{-2}$) and evaporation
$E$ (kg m$^{-2}$ s$^{-1}$):

$$Q_{0\nu} = \frac{S + 0.61\, C_p\, E}{\rho\, C_p} \tag{5}$$

with $C_p$ (=1006 J kg$^{-1}$ K$^{-1}$) the heat capacity of air, and $\rho$ the air density (kg m$^{-3}$). The friction velocity is calculated (ECMWF, 2014)
using

$$u_* = \frac{\kappa\, \sqrt{u_l^2 + v_l^2 + w_*^2}}{\ln(\frac{z_l + z_{0M}}{z_{0M}}) - \Psi_M(\frac{z_l + z_{0M}}{L}) + \Psi_M(\frac{z_{0M}}{L})} \tag{6}$$

where $u_l$ and $v_l$ are the zonal and meridional components of wind speed at 10 meters above the surface, $w_*$ is a free convection velocity
scale,

$$w_* = (z_i\, \frac{g}{T_l}\, Q_{0\nu})^{1/3} \tag{7}$$

with $z_i = 1000$ m, $z_l$ is the reference height, $z_{0H}$ and $z_{0M}$ are the roughness lengths for heat and momentum, respectively, and $\Psi_M$ is a stability
profile function for momentum (see further below).

In chemistry-transport models, the reference height $z_l$ is the altitude of the first model layer (approximately 10 m in MAGRITTE). In
comparisons with field measurements, however, the reduced reference height ($z_l = z_s - d$) should be used, with $z_s$ the sampling height, and
$d$ the zero plane displacement height, estimated as $d = 0.7 \cdot h$, with $h$ the canopy height (Karl et al., 2004). Although $R_a$ and $u_*$ are only
weakly dependent on $z_l$, it is important to use realistic values. At forest sites, $z_l$ is often of the order of 10–25 m, but it takes generally much
lower values (a few meters) in measurements setups over crops or grass, and it can be as high as 50 m in some tall-tower setups (Fowler et al.,
2011) and even higher ($> 100$ m) for determinations based on airborne observations (Cros et al., 2000).





Since the Obukhov length $L$ depends on the friction velocity (Eq. 4) which is itself dependent on $L$ (Eq. 6), an iterative method is used to solve these equations. No more than 5 iterations are needed to reach convergence.

The aerodynamic resistance (in s m$^{-1}$) is calculated using

$$R_a = \frac{1}{\kappa\, u_*} \left[ \ln\left(\frac{z_l + z_{0\mathrm{M}}}{z_{0\mathrm{H}}}\right) - \Psi_{\mathrm{H}}\left(\frac{z_l + z_{0\mathrm{M}}}{L}\right) + \Psi_{\mathrm{H}}\left(\frac{z_{0\mathrm{H}}}{L}\right) \right] \tag{8}$$

where $\Psi_{\mathrm{H}}$ is the stability profile for scalar quantities. The distributions of near-surface temperature and wind, surface sensible heat and evaporation fluxes are obtained from the ECMWF ERA-Interim operational forecasts at 3-hourly frequency on the N128 Gaussian grid. The stability profiles for heat and momentum were parameterized based on field experiments over homogeneous terrain as functions of the quantity $\zeta = \frac{z}{L}$ (Beljaars and Holtslag, 1991). In unstable conditions ($\zeta < 0$), we have

$$\Psi_{\mathrm{M}}(\zeta) = \frac{\pi}{2} - 2\arctan(x) + \ln\frac{(1+x)^2 \cdot (1+x^2)}{8} \tag{9}$$

$$\Psi_{\mathrm{H}}(\zeta) = 2\ln\left(\frac{1+x^2}{2}\right) \tag{10}$$

with $x = (1 - 16\zeta)^{1/4}$. In stable conditions ($\zeta > 0$),

$$\Psi_M(\zeta) = -b\left(\zeta - \frac{c}{d}\right)\exp(-d\zeta) - a\,\zeta - \frac{b\,c}{d} \tag{11}$$

$$\Psi_H(\zeta) = -b\left(\zeta - \frac{c}{d}\right)\exp(-d\zeta) - \left(1 + \frac{2}{3}a\zeta\right)^{1.5} - \frac{b\,c}{d} + 1 \tag{12}$$

with $a = 1$, $b = \frac{2}{3}$, $c = 5$, and $d = 0.35$.

Over ocean, the roughness lengths are calculated (ECMWF, 2014) using

$$z_{0\mathrm{M}} = 0.11\,\frac{\nu}{u_*} + \alpha_{\mathrm{Ch}}\,\frac{u_*^2}{g} \tag{13}$$

$$z_{0\mathrm{H}} = 0.4\,\frac{\nu}{u_*} \tag{14}$$

where $\nu$ is the kinematic viscosity and $\alpha_{\mathrm{Ch}}$ the dimensionless Charnock coefficient (Charnock, 1955) also provided by ERA-Interim. The kinematic viscosity (= $1.5 \cdot 10^{-5}$ m$^2$ s$^{-1}$ at 288 K) is calculated (Nobel, 1983) using

$$\nu = \frac{a_1\,T_l^{1.5}}{\rho\,(T_l + \mathrm{Su})} \tag{15}$$

with $a_1 = 1.458 \cdot 10^{-6}$ kg m$^{-1}$ s$^{-1}$ K$^{-1/2}$ and Su = 110.4 K. Over land, minimum and maximum values of $z_{0\mathrm{M}}$ are defined for 8 plant
functional types (Table 3.2), for which we use the geographical distribution (at $0.5° \times 0.5°$ resolution) of the MEGAN model (Guenther et al., 2006). $z_{0\mathrm{H}}$ is taken equal to $\frac{z_{0\mathrm{M}}}{10}$ over land (ECMWF, 2014). Following Zhang et al. (2003), the seasonal evolution of $z_{0\mathrm{M}}$ and $z_{0\mathrm{H}}$ follows the leaf area index (LAI), i.e.

$$z_0(t) = z_0^{\min} + (z_0^{\max} - z_0^{\min}) \times \frac{\mathrm{LAI}(t) - \mathrm{LAI}^{\min}}{\mathrm{LAI}^{\max} - \mathrm{LAI}^{\min}} \tag{16}$$

where LAI is the monthly averaged LAI from the MODIS MOD15A2 composite product (obtained from http://reverb.echo.nasa.gov/), and
LAI$^{\min}$ and LAI$^{\max}$ are the minimum and maximum monthly LAI over the course of the year.



**Table 2.** Plant functional types (PFT) and values of parameters involved in the dry deposition scheme. All resistance units are s m$^{-1}$.

| | $z_{0M}^{\min}$ | $z_{0M}^{\max}$ | $a_s$ | $b_s$ | $c_s$ | $d_s$ | $R_{ac0}^{\min}$ | $R_{ac0}^{\max}$ | $R_{cutd0}^{O_3}$ | $R_{cutw0}^{O_3}$ | $R_{cutd0}^{SO_2}$ | $R_{gd}^{SO_2}$ |
|---|---|---|---|---|---|---|---|---|---|---|---|---|
| | m | m | J m$^{-3}$ | J m$^{-2}$ | | hPa$^{-1}$ | | | | | | |
| Needleleaf evergreen trees | 2.0 | 2.0 | 14000 | 2 | 18 | 0.031 | 100 | 100 | 6000 | 300 | 2000 | 200 |
| Needleleaf deciduous trees | 2.0 | 0.4 | 14000 | 2 | 18 | 0.031 | 60 | 100 | 6000 | 300 | 2000 | 200 |
| Broadleaf evergreen trees | 2.5 | 2.5 | 4000 | 6 | 20 | 0.036 | 250 | 250 | 9000 | 600 | 2500 | 100 |
| Broadleaf deciduous trees | 1.0 | 0.4 | 10000 | 5 | 30 | 0.036 | 100 | 250 | 9000 | 600 | 2500 | 200 |
| Shrub | 0.2 | 0.05 | 10000 | 5 | 30 | 0.031 | 60 | 60 | 7500 | 450 | 2000 | 200 |
| Crop | 0.05 | 0.02 | 20000 | 6 | 90 | 0.0 | 10 | 40 | 6000 | 300 | 1500 | 200 |
| Grass | 0.05 | 0.05 | 5000 | 1 | 50 | 0.024 | 10 | 40 | 6000 | 300 | 1000 | 200 |
| Desert | 0.005 | 0.005 | – | – | – | – | – | – | – | – | – | – |

## 3.2 Quasi-laminar sublayer resistance, $R_b$

The quasi-laminar sublayer resistance of a compound $i$ ($R_{b,i}$, s m$^{-1}$) is expressed (Toyota et al., 2016) as

$$R_{b,i} = \frac{1}{B \, u_*} \left( \frac{\nu}{\mathrm{Pr} \, D_{g,i}} \right)^{2/3} \tag{17}$$

where $B$ is an empirical constant taken equal to $\frac{\kappa}{2}$ (although this approximation is not very accurate over sparse vegetation and water),

Pr=0.72 is Prandtl's number, and $D_{g,i}$ is gas-phase diffusivity. The diffusivities (in m$^2$ s$^{-1}$) are based on the parameterization of Fuller et al. (1966, 1969):

$$D_{g,i} = \frac{10^{-7} \, T_l^{1.75} \, (1/\mathrm{MW}_i + 1/\mathrm{MW}_{\mathrm{air}})^{1/2}}{p \, (v_i^{1/3} + v_{\mathrm{air}}^{1/3})^2} \tag{18}$$

where $\mathrm{MW}_i$ and $\mathrm{MW}_{\mathrm{air}}$ denote the molecular weights (g mol$^{-1}$) of compound $i$ and of ambient air, $p$ is pressure (atm), and $v_i$ is the "diffusion volume" (dimensionless) of compound $i$, calculated as a sum of atomic diffusion volumes, with each C, H, O, N, Cl and S atom

contributing 15.9, 2.31, 6.11, 4.54, 21 and 22.9, respectively, whereas aromatic or heterocyclic rings contribute for -18.3. The diffusion volume of air molecules is $v_{\mathrm{air}}$ = 19.7 cm$^3$. Equation Eq. 18 gives predictions very close to experiment-based estimates (typically within 10%) for many organic compounds including alkanes, alkenes, aromatics as well as monofunctional ketones, alcohols and carboxylic acids (Tang et al., 2015). The values of the diffusion coefficients at 298 K used in this study for oxygenated organic compounds are given in Table 1. When available, experiment-based estimates are used (Tang et al., 2014, 2015; Massman, 1998) with the temperature- and pressure-

dependence of Eq. 18. Note that $D_g$ at 298 K and 1 atm is estimated at 0.251 cm$^2$ s$^{-1}$ for H$_2$O, 0.151 cm$^2$ s$^{-1}$ for HNO$_3$, 0.154 cm$^2$ s$^{-1}$ for H$_2$O$_2$, 0.176 cm$^2$ s$^{-1}$ for O$_3$, and 0.125 cm$^2$ s$^{-1}$ for SO$_2$.

## 3.3 Surface resistance, $R_c$

The determination of the bulk surface resistance follows a resistance analogy formulation (Wesely, 1989). Following Zhang et al. (2003), it is written as

$$\frac{1}{R_c} = \frac{1}{R_s + R_m} + \frac{1}{R_{ac} + R_g} + \frac{1}{R_{cut}} \tag{19}$$





where $R_s$ is stomatal resistance, $R_m$ is mesophyll resistance, $R_g$ is the resistance to soil uptake, $R_{\text{cut}}$ is the cuticular resistance, and $R_{\text{ac}}$ the resistance to transfer in the canopy. Over oceans, $R_c = R_g$.

### 3.3.1 Surface resistance over ocean

Following Liss and Slater (1974), the net air–sea flux ($F$) of a chemical compound is the difference between gross oceanic uptake ($U$) and gross oceanic emission ($E$), and can be written as the product of an exchange coefficient ($K_g$) by the difference in concentration between the air ($C_g$) and the water ($C_l$):

$$F = U - E = K_g \left( C_g - \frac{C_l}{H} \right) \tag{20}$$

where $H$ is the dimensionless Henry's law constant, related to the HLC ($K_H$) defined in Sect. 2 by

$$H = R \cdot T_s \cdot K_H \tag{21}$$

with $R$ the gas constant (= 0.08206 L atm K$^{-1}$ mol$^{-1}$) and $T_s$ the temperature at the surface (Sander, 2015). $K_g$ is expressed as

$$\frac{1}{K_g} = \frac{1}{k_g} + \frac{1}{H \cdot k_l}, \tag{22}$$

where $k_g$ and $k_l$ represent the gas-phase and liquid transfer velocities, respectively. A positive $F$ corresponds here to a net oceanic uptake. Estimates are lacking for the oceanic subsurface concentration $C_l$ of most compounds, except methanol ($C_l = 118$ nmol l$^{-1}$) (Williams et al., 2004), acetone (15 nmol l$^{-1}$) (Fischer et al., 2012) and acetaldehyde ($C_l$ distribution parameterized as in Millet et al. (2010)), for which oceanic emissions are implemented in the model. For other OVOCs, those emissions are neglected. This assumption leads to a probable overestimation of the net oceanic uptake; more work is needed to assess the significance of OVOC oceanic emissions.

Although numerous parameterizations exist for $k_g$ (see Johnson (2010) and references therein), its inverse can be identified with the sum of the aerodynamic and quasi-laminar layer resistances ($R_a + R_b$) discussed above. The distribution of $1/(R_a + R_b)$ parameterized as described in Sec. 3.1-3.2 is shown on Fig. 1 and compared with the gas-phase transfer velocity parameterization of Johnson (2010). The two distributions are very similar, with differences generally well below 20%. The transfer velocity determined according to Johnson is on average ca. 3% higher than the corresponding velocity parameterized in accordance with ECMWF IFS formulation.

The surface resistance to deposition onto water surfaces can be written as

$$R_c = ( H \cdot k_l + f_0 / R_{w0} )^{-1} \tag{23}$$

with $R_{w0}$ (= 2000 s m$^{-1}$) the resistance of water surfaces to the reactive uptake of ozone (Wesely, 1989). Whereas the first term in the rhs of Eq. 23 accounts for the solubility of the compound in the ocean mixed layer, the second term accounts for its reactive uptake at the air-sea interface. This term is usually very minor, but it is the dominant deposition pathway for highly reactive but poorly soluble compounds like ozone. The values of the $f_0$ parameter (between 0 and 1), adapted from Wesely (1989), are given in Table 1.

The liquid transfer velocity $k_l$ is parameterized according to Nightingale et al. (2000), as described in Johnson (2010):

$$k_l = (0.222\,U_{10}^2 + 0.333\,U_{10}) \cdot \left( \frac{S_{cw}}{600} \right)^{-0.5} \tag{24}$$

where $U_{10}$ is the wind speed at 10 m above the surface and $S_{cw}$ is the Schmidt number calculated according to

$$S_{cw} = \frac{\eta_w}{\rho_w D_{w,i}}, \tag{25}$$



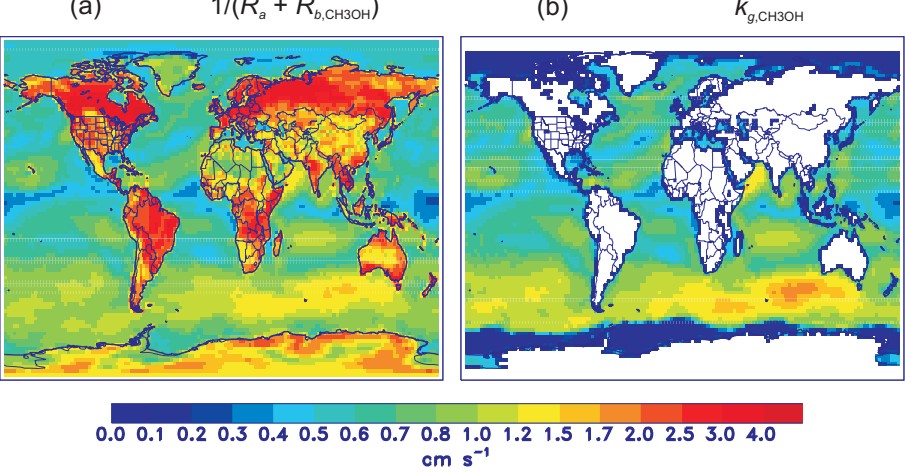

**Figure 1.** (a) Inverse of the sum of the aerodynamic resistance and quasi-laminar boundary layer resistance $1/(R_a + R_b)$ for methanol, and (b) gas-phase air-sea transfer velocity for methanol according to Johnson (2010), both for the month of July.

with $\eta_w$ the dynamic viscosity of pure water (kg m$^{-1}$ s$^{-1}$), well approximated by

$$\eta_w = \frac{T_C + 246}{0.05594 \cdot T_C^2 + 5.2842 \cdot T_C + 137.37} \tag{26}$$

where $T_C$ is temperature in °C, $\rho_w$ the density of seawater, very close to 1025 kg m$^{-3}$ at a typical salinity ($S$) of 35 g kg$^{-1}$, and $D_{w,i}$ the diffusivity of compound $i$ in water (m$^2$ s$^{-1}$). The latter (of the order of $10^{-9}$ m$^2$ s$^{-1}$) is calculated using

$$5 \quad D_{w,i} = \frac{5.1 \cdot 10^{-14} T}{\eta_s V_{b,i}^{0.6}} \tag{27}$$

where $\eta_s$ is the dynamic viscosity of seawater (kg m$^{-1}$ s$^{-1}$) and $V_{b,i}$ is the liquid molar volume (cm$^3$ mol$^{-1}$) of the compound. The temperature dependence of $\eta_s$ is obtained by interpolation of experimental values at $S$=35 g kg$^{-1}$ reported by Johnson (2010): $\eta_w = 2.244$, 1.880, 1.605, 1.390, 1.219, 1.08, 0.965, 0.870, 0.786 $\times 10^{-3}$ kg m$^{-1}$ s$^{-1}$ at temperatures between -5 and 35°C by increments of 5°C. The liquid molar volumes are estimated using Schroeder's additive method (Partington, 1949), according to which each C, H, O, N atom contributes for 7 cm$^3$ mol$^{-1}$, whereas S atoms, rings and double bonds contribute for 21, -7 and 7 cm$^3$ mol$^{-1}$, respectively.

### 3.3.2 Deposition through plant stomata

The stomata are actively regulated openings of the epidermis of leaves. They regulate the exchange of $CO_2$ and $H_2O$ between the plant and the air. The stomatal resistance ($R_s$) is highly variable and depends on meteorological conditions and on the leaf water potential. The mesophyll resistance ($R_m$) is an additional, species-specific resistance involved in the deposition through the plant stomata (Eq. 19). It is negligibly low for ozone and for highly soluble compounds, but can be significant for less reactive, poorly soluble compounds (Wesely, 1989). However, eddy covariance and gradient measurements of OVOC fluxes at forested sites have revealed high deposition velocities (Karl et al., 2010) suggesting low mesophyll resistances even for poorly soluble species such as MVK. The specification of $R_m$ for all compounds is described in Sect. 3.3.4.





Since temperature and light fluxes are variable within the canopy, the stomatal conductance of the canopy is calculated using a multi-layer canopy environment model, MOHYCAN (Müller et al., 2008) by combining the contributions of sunlit and shaded leaves at every layer in the canopy:

$$\frac{1}{R_s} = \sum_{k=1}^{n} \left[ \frac{\text{LAI}_{\text{sun}}^k}{r_s^{\text{sun},k}} + \frac{\text{LAI}_{\text{shade}}^k}{r_s^{\text{shade},k}} \right] \cdot f_l \tag{28}$$

where $n$ (= 8) is the number of layers, $\text{LAI}_{\text{sun}}^k$ and $\text{LAI}_{\text{shade}}^k$ are the leaf area index of sunlit and shaded leaves at layer $k$, $f_l$ is the effective number of sides of the leaves (equal to 1 for shrub and to 1.25 for other PFTs) and $r_s^{\text{sun},k}$ and $r_s^{\text{shade},k}$ are the stomatal resistances of sunlit and shaded leaves at layer $k$, respectively, parameterized according to the Simple Biosphere model (SiB) (Sellers et al., 1986) based on previous work (Jarvis, 1976):

$$r_s = \frac{r_s^{\text{op}}(Q)}{f(T_l) \cdot f(\delta_e) \cdot f(\psi_l)} \cdot \frac{D_{g,\text{H2O}}}{D_{g,i}} \tag{29}$$

where $r_s^{\text{op}}(Q)$ is the optimal stomatal resistance for $H_2O$ in unstressed conditions, depending on the photosynthetic photon flux density ($Q$, in W m$^{-2}$), and the $f$ functions are stress factors for temperature ($T_l$), water vapour deficit ($\delta_e$ in hPa) and leaf water potential ($\psi_l$). $r_s^{\text{op}}(Q)$ and the stress functions are evaluated separately for sunlit and shaded leaves at each canopy layer. These stress functions are detailed in Müller et al. (2008). $D_{g,\text{H2O}}$ and $D_{g,i}$ are the gas-phase diffusivities of water vapour and the compound under consideration, calculated as in Sect. 3.2. The optimal resistance $r_s^{\text{op}}(Q)$ depends primarily on the visible radiation flux:

$$r_s^{\text{op}}(Q) = \frac{a_s}{b_s + Q} + c_s \tag{30}$$

where $a_s$, $b_s$ and $c_s$ are given in Table 3.2. The water deficit stress function is expressed as

$$f(\delta_e) = 1 - d_s \, \delta_e \tag{31}$$

with $d_s$ also given in Table 3.2 . The values of the resistance parameters originally used in the SiB model (see Müller et al.) led to overestimated stomatal resistances, and underestimated ozone deposition velocities for broadleaf (deciduous or evergreen) forests (Val Martin et al.,
2014). The values of $a_s$, $b_s$ and $c_s$ shown on Table 3.2 were therefore adjusted on the basis of comparisons with ozone deposition velocity measurements in various environments (Sect. 3.4 and Table 3.4). Furthermore, the $d_s$ parameter for broadleaf evergreen forest was also increased (to the same value as for broadleaf deciduous trees) based on the strong dependence of ozone deposition velocity on water vapour deficit observed over Amazonian forests, as discussed in Sect. 3.4. This adjustment of stomatal resistances is justified by the dominance of stomatal uptake for ozone during daytime (Zhang et al., 2006), except over sparsely vegetated areas. The canopy stomatal resistance ($R_s$) is
minimum around noon in summertime. It is typically of the order of 100 s m$^{-1}$ for broadleaf deciduous trees, in agreement with observations (Baldocchi et al., 1987; Padro, 1996; Val Martin et al., 2014). For broadleaf evergreen forests, even lower stomatal resistances (often of the order of 50 s m$^{-1}$ or less) are suggested by the high deposition velocities of ozone and several OVOCs measured in tropical rainforests (see further below).

The crudeness of the above PFT-based approach should be acknowledged: there are known very large interspecies differences within a
given PFT, e.g. up to a factor of 5 between the stomatal resistances of different crop species, or a factor of 2–3 between several broadleaf deciduous trees (Baldocchi et al., 1987; Meyers et al., 1998). This limitation should be kept in mind in model comparisons against field measurements.





### 3.3.3 Cuticle and ground deposition of $O_3$ and $SO_2$

The deposition of $O_3$ and $SO_2$ to the cuticle is favoured by rain or dew. The calculation of cuticular resistance therefore distinguishes wet and dry canopies (Zhang et al., 2003):

$$\frac{1}{R_{\text{cut}}} = \frac{1 - f_{\text{wet}}}{R_{\text{cutd}}} + \frac{f_{\text{wet}}}{R_{\text{cutw}}} \tag{32}$$

where $f_{\text{wet}}$ is the frequency of rain or dew conditions, and

$$R_{\text{cutd}} = \frac{f_{\text{freez}}(T_C) \cdot R_{\text{cutd0}}}{e^{0.03\,\text{RH}}\,\text{LAI}^{0.25}\,u_*} \tag{33}$$

$$R_{\text{cutw}} = \frac{f_{\text{freez}}(T_C) \cdot R_{\text{cutw0}}}{\text{LAI}^{0.5}\,u_*} \tag{34}$$

where RH is relative humidity (%), and $R_{\text{cutd0}}$ and $R_{\text{cutw0}}$ are reference values for dry and wet cuticle resistances, respectively. Their values

for $O_3$, and the values of $R_{\text{cutd0}}$ for $SO_2$ are given in Table 3.2 for the different PFTs; in addition, $R_{\text{cutw0}}^{SO_2}$ is taken to be 50 and 100 s m$^{-1}$ for rain and dew conditions, respectively. The function $f_{\text{freez}}(T_C)$ is equal to 1 above –1°C, and is given by

$$f_{\text{freez}}(T_C) = \min(\,2\,, e^{0.2(-1-T_C)}\,) \tag{35}$$

below –1°C.

The frequency of rain is determined from ECMWF cloud and precipitation fields as described in Stavrakou et al. (2009b). Dew occurs

when the friction velocity $u_*$ does not exceed a critical value (Janssen and Romer, 1991; Brook et al., 1999)

$$u_* < \frac{1.5 \cdot C_0}{q_s(T) - q_s(T_d)} \tag{36}$$

where $q_s(T)$ and $q_s(T_d)$ are the saturated specific humidity at ambient temperature $T$ and at dew temperature $T_d$, and $C_0$ is a constant equal to 0.3 for a cloud fraction (CC) lower than 0.25, 0.2 for CC between 0.25 and 0.75, and 0.1 for CC above 0.75.

The resistance to the transfer in the canopy ($R_{\text{ac}}$) is identical for all compounds and parameterized (Zhang et al., 2003) using

$$R_{\text{ac}} = \frac{R_{\text{ac0}}\,\text{LAI}^{0.25}}{u_*^2} \tag{37}$$

where $R_{\text{ac0}}$ varies seasonally between PFT-specific mininum and maximum values ($R_{\text{ac}}^{\min}$ and $R_{\text{ac}}^{\max}$) given in Table 3.2. The seasonal evolution of $R_{\text{ac0}}$ is assumed to follow the LAI, similarly to the roughness length (see above, Eq. 16).

The resistance to deposition of ozone to the ground ($R_g^{O_3}$) is taken equal to 200 or 500 s m$^{-1}$ for vegetated and non-vegetated surfaces, respectively (Zhang et al., 2003). In cold conditions, this resistance is enhanced by the factor $f_{\text{freez}}$ defined in Eq. 35. For $SO_2$, the

ground resistance ($R_g^{SO_2}$) is taken equal to 50 and 100 s m$^{-1}$ for rain and dew conditions, respectively, and is also multiplied by $f_{\text{freez}}$. In absence of dew or rain, the resistance depends on soil pH and relative humidity (Ganzeveld et al., 1998; Kerkweg et al., 2006). For sufficiently humid conditions (RH > 60%), the resistance ($R_{g,\text{humid}}^{SO_2}$) is equal to 115, 65 or 25 s m$^{-1}$ for pH < 5.5, 5.5 < pH < 7.3 and pH > 7.3, respectively. The distribution of soil pH at $0.5° \times 0.5°$ resolution is obtained from the SoilGrids database (Hengl et al., 2017) (ftp://ftp.soilgrids.org/data/aggregated/). At lower RH, $R_g^{SO_2}$ is modified as follows:

$$R_g^{SO_2} = 3.4 \cdot R_{g,\text{humid}}^{SO_2} - 85 + 10^5 \cdot \max(0, (40 - \text{RH})/40) \tag{38}$$





In cold conditions, $R_g^{\mathrm{SO_2}}$ is enhanced by adding a contribution $1000 \cdot e^{269-T_s}$, with $T_s$ being the soil temperature in K. To avoid very low values over deserts, a minimum value of 25 s m$^{-1}$ is imposed to $R_g^{\mathrm{SO_2}}$. Over sea ice, a separate formulation is used (Zhang et al., 2003):

$$R_g^{\mathrm{SO_2}} = \min(500, \max(100, 70(2 - T_C)))$$
(39)

Finally, over snow, both surface and cuticular resistances for $O_3$ are taken equal to 2000 s m$^{-1}$, and the surface resistance for $SO_2$ is
calculated (Kerkweg et al., 2006) using

$$R_{c,\mathrm{snow}}^{\mathrm{SO_2}} = \min(10^5, 10^{-0.09\,T_C + 2.4})$$
(40)

The snow cover fraction ($f_{\mathrm{snow}}$) is estimated from the ECMWF snow depth (SD) analysis. Similar to Zhang et al. (2003), it is expressed as a ratio

$$f_{\mathrm{snow}} = \frac{\mathrm{SD}}{\mathrm{SD}_{\mathrm{max}}}$$
(41)

where $\mathrm{SD}_{\mathrm{max}}$ (m) is equal to $\max(0.2 \cdot \mathrm{LAI}, 0.02)$.

### 3.3.4   Canopy resistance to OVOC deposition

The calculation of the mesophyll, ground and cuticular resistances for any chemical compound is adapted from Wesely (1989) and Zhang et al. (2002). The conductances are expressed as linear combinations of the corresponding conductances for $SO_2$ (as template for water-soluble compounds) and $O_3$ (template for very reactive compounds):

$$\frac{1}{R_m} = \frac{K_H \cdot f_1}{3000} + 100 \cdot f_0$$
(42)

$$\frac{1}{R_g} = \frac{K_H \cdot f_1}{10^5 \cdot R_g^{\mathrm{SO_2}}} + \frac{f_0}{R_g^{O_3}}$$
(43)

$$\frac{1}{R_{\mathrm{cut}}} = \frac{K_H \cdot f_1}{10^5 \cdot R_{\mathrm{cut}}^{\mathrm{SO_2}}} + \frac{f_0}{R_{\mathrm{cut}}^{O_3}}$$
(44)

where $f_0$ and $f_1$ are empirical, species-dependent factors crudely estimated from comparisons with available field measurements (see Sect. 3.5). $f_0 = 0$ for non-reactive species, whereas $f_0 = 1$ for species as reactive as ozone. The $f_0$ and $f_1$ values adopted in this work are provided in Table 1. In the original model of Wesely, $f_1$ was omitted, i.e. $f_1 = 1$. The model of Zhang et al. (2002) has a similar formulation with two species-depdendent factors. For example, the ground resistance in their model (also in Paulot et al. (2018)) is expressed as

$$\frac{1}{R_g^{\mathrm{Z02}}} = \frac{\alpha}{R_g^{\mathrm{SO_2}}} + \frac{\beta}{R_g^{O_3}}$$
(45)

i.e. $\beta$ can be identified with our $f_0$, and $\alpha$ is related to $f_1$ by

$$\alpha = f_1 \cdot \frac{K_H}{10^5}$$
(46)

However, the values adopted by Zhang et al. (2002) for $\alpha$ were poorly constrained, and taken equal to zero for the least soluble compounds like MVK, MACR, PAN and $CH_3CHO$. Paulot et al. (2018) provided updated estimates for $\alpha$ and $\beta$ for several compounds, with a focus





on organic nitrates, based on flux measurements for a suite of OVOCs in Alabama (Nguyen et al., 2015). The values of $\alpha$ at 298 K from our work are also provided in Table 1, to facilitate comparison with their work (see further below).

As discussed in the next Section, $f_1$ values frequently much higher than unity are found necessary to bring the model in agreement with available measurements for many OVOCs. Note that in those cases, the precise value of $f_0$ (assumed to lie in the range from 0 to 1) is most

often unimportant, because the solubility-related term in the right-hand side of Eq. (42)-(44) becomes largely dominant. For simplicity, we assumed $f_0 = 0$ for all compounds, except the hydroperoxides and peracids ($f_0 = 0.1$) as well as the PAN-like compounds ($f_0 = 1$) (Table 1). Different values could have been chosen for $f_0$, but with generally very little consequences for the predicted deposition velocities, except in a few cases (e.g. HCHO and PAN). Note that Karl et al. (2010) recommended to take $f_0 = 1$ as a rule for most oxygenated VOCs.

### 3.4 Dry deposition model evaluation for ozone and sulfur dioxide

Tables 3.4 and 3.4 list the $O_3$ and $SO_2$ deposition measurement campaigns used for evaluation. For each campaign, the meteorology is provided by ECMWF ERA-Interim operational forecasts for the year and month(s) of the measurements. The calculations use the reported LAI and plant functional type, when available. The model time step is 1 hour. The reference height ($z_l$) used in the calculations is obtained from the reported sampling and canopy heights. When not reported, $z_l$ is taken to be 3 m for crop and grassland sites, and 15 m for forest sites.

In the case of ozone, every major PFT except shrub is represented by at least 4 campaigns in Table 3.4. For each PFT, the model performs well on average, although large discrepancies are found at specific sites. The highest ozone deposition velocities ($v_{d,O_3}$) are found in tropical rainforests (1.06 cm s$^{-1}$ on average for the measurements, vs. 1.16 cm s$^{-1}$ according to the model), followed by broadleaf deciduous forests (0.51 vs. 0.48 cm s$^{-1}$), needleleaf forests (0.39 vs. 0.41 cm s$^{-1}$), crops (0.36 vs. 0.39 cm s$^{-1}$), and grasslands (0.27 vs. 0.31 cm s$^{-1}$). As seen on Fig. 2–4, the observed diurnal cycle of $v_{d,O_3}$ is generally well reproduced by the model, with highest values found during daytime,

primarily due to the strong response of stomatal resistances to solar radiation (Eq. 29–30). Exceptions include the cases of Harvard forest in September/October (Fig. 2(a)) and Rebio Jarú in Rondônia during the dry season (September/October) (Fig. 3(a)), where the model fails to match the unexpectedly high values measured during late night (3–6 AM local time). As discussed by Wu et al. (2011), the high values at Harvard forest might be due to mixing/transport events not well represented by the resistance analogy; furthermore, they are based on only few measurements and might not be typical. Non-stomatal uptake by leaves was presumed to explain the high nighttime deposition at Rebio

Jarú during the dry season (Rummel et al., 2007). Reaction with nitric oxide emitted by soil could also contribute, but was discarded as major cause for the departure from model expectation (Rummel et al., 2007).

In contrast, the model overestimates the late-night deposition velocities at a rainforest site (Bukit Atur) in Borneo (Fig. 3(b)), indicating a large underestimation of the aerodynamic resistance (only about 100–200 s m$^{-1}$ in the simulation at 5-6 LT). The forest canopy was shown to be isolated from the boundary layer air due to a nocturnal temperature inversion (Fowler et al., 2011). This issue is likely related to the

complex terrain at the site, with the tower being located on a hill 260 m above the valley bottom. The daily course of the sensible heat flux in the model, maximum at about 70 W m$^{-2}$ at midday and slightly negative during the night, is in fair agreement with the observations (Fowler et al., 2011), as is also the friction velocity (on which $R_a$ and $R_b$ are strongly dependent), of the order of 0.1–0.15 cm s$^{-1}$ during the night in both the model and the measurements (Langford et al., 2010).

The deposition velocities show important seasonal variations due to their dependence on meteorological variables, leaf water potential, and leaf area index. The observed $v_{d,O_3}$ values at an oak forest site in Italy (Fig. 2(c)) are higher during spring than during summer (by

approximately 50%), in good agreement with the model simulations. This reflects the influence of the stress factors (mainly $f(\delta_e)$ and $f(\psi_l)$, see Eq. 29) on stomatal uptake, which was found to dominate overall deposition at the site (Fares et al., 2014). The effect of the higher solar





**Figure 2.** Average measured (symbols) and modelled (curves) ozone deposition velocities (cm s$^{-1}$) at temperate and boreal forest sites (see Table 3.4). At Hyytiälä (panel (e)), the crosses and stars represent the reported measurements (Rannik et al., 2012) above and below 70% relative humidity, respectively (average relative humidity was below that threshold between 10 and 20 LT).

radiation levels during summer is more than compensated by the higher vapour pressure deficit, by almost a factor 2 (Fares et al., 2014), and by the lower soil water content resulting in a lower value of the leaf water potential stress factor $f(\psi_l)$, by a factor of 1.7 according to the model simulation. At a tropical grassland site in Rondônia as well (Fig. 4), the modelled and measured $v_{d,O_3}$ values are consistently higher





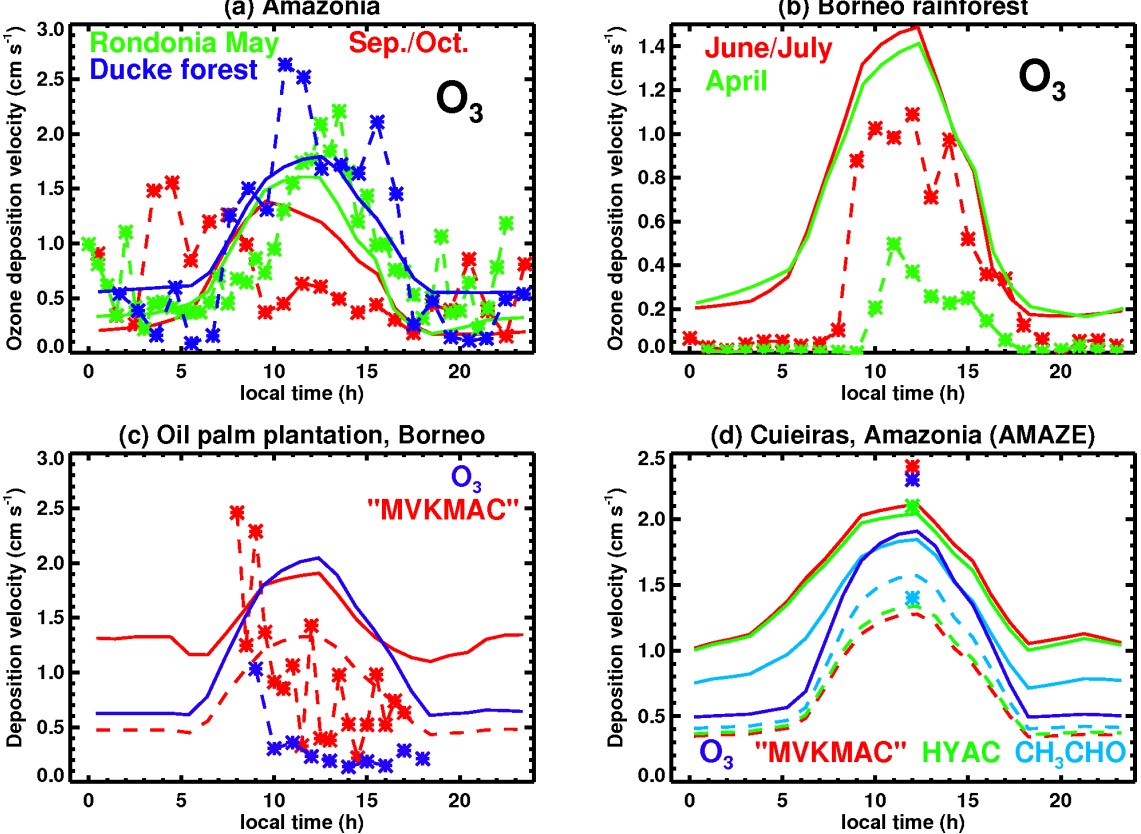

**Figure 3.** Average measured (symbols) and modelled (curves) deposition velocities (cm s$^{-1}$) at tropical forest sites for (a)-(b) ozone (see Table 3.4) and (c)-(d) ozone and other compounds (Table 5). "MVKMAC" denotes the sum MVK + MACR corrected for the interference due to ISOPOOH (see text). The dashed curves for the OVOCs correspond to results obtained with $f_0 = f_1 = 1$ in the calculation of the canopy resistance.

during the wet season (by about 50%) due to the combination of lower vapour pressure deficit (hence higher $f(\delta_e)$), higher soil moisture (higher $f(\psi_l)$ by a factor 1.4), and higher LAI (2.1 in May and 1.2 in October, see Table 3.4).

At the tropical forest sites (Fig. 3(a)–(b)), however, very large variations in daytime deposition velocities are observed, which the model is unable to reproduce. Three different campaigns in Amazonia during the wet season indicate midday $v_{d,O_3}$ close to 2 cm s$^{-1}$, suggesting
5   canopy resistances of the order of 30–40 s m$^{-1}$ in optimal (unstressed) conditions. Note that a fraction of the observed conductance is due to in-canopy reactive loss of ozone, dominated (for more than 50%, Yee et al. (2018)) by the reaction with sesquiterpenes released by vegetation and soils; however, the contribution of sesquiterpenes should not exceed ca. 0.1 cm s$^{-1}$ based on reported fluxes (Jardine et al., 2011a; Bourtsoukidis et al., 2018). During the dry season at the Rebio Jarú site in Rondônia, the deposition velocity dropped by about a factor of three compared with the wet season, whereas the model-calculated $v_{d,O_3}$ was only about 30% lower (Fig. 3(a)). The averaged leaf water
10   potential stress factor $f(\psi_l)$ differed by only a factor of 1.25 between the two seasons at Rebio Jarú according to our parameterization based on ECMWF ERA-Interim fields; at Bukit Atur, it is predicted to be constant all year round. However, drought was clearly responsible for the




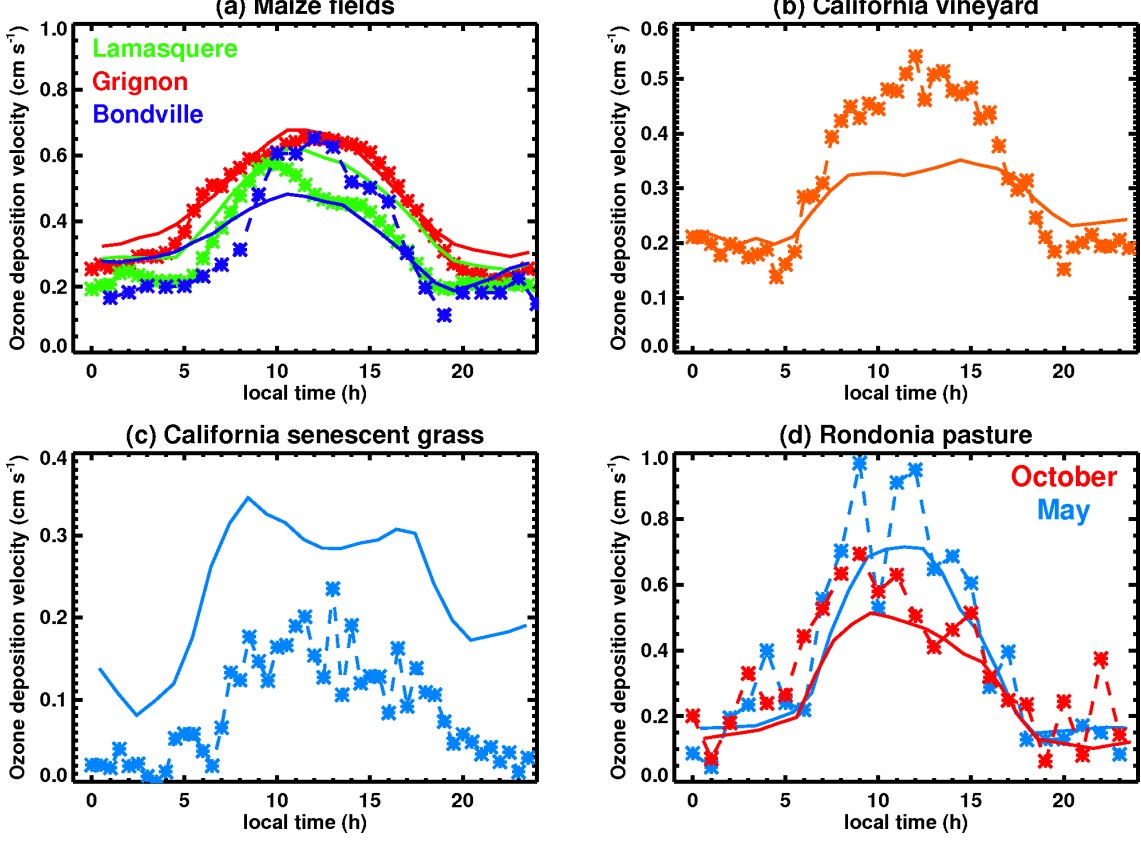

**Figure 4.** Average measured (symbols) and modelled (curves) ozone deposition velocities (cm s$^{-1}$) at (a)-(b) crop and (c)-(d) grassland sites (see Table 3.4).

$v_{d,O_3}$ shutdown in Rondônia in September/October. An analysis of the dry season fluxes by Rummel et al. (2007) indicated that high canopy resistances, of the order of 300 s m$^{-1}$, were associated with high values of the water vapour deficit ($\delta_e > 16$ hPa), whereas the resistance was about three times lower in more humid/cool conditions ($\delta < 16$ hPa). This suggests that to a large extent, the seasonality of $v_{d,O_3}$ is controlled by changes in the associated stress factor $f(\delta_e)$. The measured average values of the H$_2$O pressure deficit at midday, ca. 13 and

5    22 hPa in the wet and the dry season, respectively (Rummel et al., 2007), imply higher stomatal resistances during the dry season, by about a factor 2.5 according to Eq. 31 (with $d_s = 0.036$ hPa$^{-1}$). If correct, and taking into account the other model-calculated stress factors and the midday value of $R_a + R_b$ estimated from the measurements (ca. 25 s m$^{-1}$) (Rummel et al., 2007), this would translate into a factor of almost three between dry and wet season $v_{d,O_3}$, in excellent consistency with the observed deposition velocities. The relatively poor performance of the ECMWF-driven model displayed on Fig. 3 is mainly due to a significant cold bias of the midday ERA-Interim temperatures at this

10   site, by about 3 K in May and as much as ca. 5 K in September/October 1999. Although other factors might be at play, the underestimated vapour pressure deficit induced by this cold bias is the most likely explanation for the model discrepancy.

At Bukit Atur in Borneo, however, the measured meteorological conditions were quite similar in April (late wet season) and in June/July (early dry season) (Langford et al., 2010), and they were well reproduced by the ECMWF analysis at this site (Stavrakou et al., 2014). The





large variation of $v_{d,O_3}$ between the two periods is therefore difficult to explain. The friction velocity did not show important long-term variations (Langford et al., 2010). The soil water stress factor $f(\psi_l)$ could possibly be more variable than estimated here. Phenological changes might also play an important role. More field measurements are needed to better characterize the long-term changes of deposition velocities in tropical ecosystems.

The choice of the stomatal resistance parameters (Table 3.2), largely based on the evaluation of modelled $v_{d,O_3}$ and on previous literature estimates, could be biased due to the limited representativity of the sites at which measurements are available. This is especially true for crops, and probably also for the other PFTs. Note that the stomatal resistance parameters for shrub were assumed similar to those of broadleaf deciduous trees, due to the lack of field data over shrub. Besides the large differences between the stomatal resistances of different species (within a given PFT), many factors might contribute to model errors, including the limited reliability and representativity of

(micro-)meteorological fields based on ECMWF analyses, the poorly understood role of surface wetness (Zhang et al., 2002), the still crude representation of cuticular and ground resistances, and the poorly quantified role of in-canopy chemical processing. As for $O_3$, the model performs well on average for $SO_2$ deposition velocities (Table 3.4), with measured and modelled average deposition velocities of 0.61 and 0.64 cm s$^{-1}$, respectively. Unfortunately, the modelled $v_{d,SO_2}$ could not be evaluated for tropical ecosystems, due to the lack of field data in those environments.

## 3.5   Dry deposition model evaluation for OVOCs

Whereas the relationship between the stomatal resistances of different compounds (Sect. 3.3.2) is relatively straightforward (under the reasonable assumption that their diffusivities are well approximated), the other critical resistances involved in the canopy conductance are much more variable, not well understood and probably much less well described by current deposition models. In the original formulation of Wesely (1989), the reactivity factor $f_0$ (Sect. 3.3.4) was taken equal to 0 or 0.1 for all species but ozone, based on estimated electron

activity for halfredox reactions in aqueous neutral solutions and on second-order reaction rates with S(IV). These values (combined with the assumption $f_1 = 1$) imply very low deposition velocities (at most a few mm s$^{-1}$) for almost all organic compounds. Based on limited data, higher $f_0$ values were adopted by Zhang et al. (2002), e.g. 0.6 for PAN, 0.8 for organic hydroperoxides and 0.5 for organic nitrates, but still only 0.05 for most carbonyls, and their values for the scaling factor of the solubility-related term of the conductances ($f_1$ or $\alpha$, Eq. 42–46) were very low except for the most soluble compounds.

More recently, the measurement over temperate and tropical forests of very high (and very similar) midday deposition velocities, of the order of 2 cm s$^{-1}$, for the poorly soluble MVK + MACR as well as for the more soluble GLYALD and HYAC (Karl et al., 2010) indicates that solubility plays only a minor role in determining the conductance of OVOCs, at least for the range of compounds considered. In order to rationalize their similar deposition velocities and their similarity with ozone, Karl et al. (2010) proposed to assume $f_0 = 1$, resulting in similar deposition velocities for all compounds.

However, this view has been challenged by deposition measurements at a deciduous forest site in Brent, Alabama (Nguyen et al., 2015) suggesting a strong relationship between solubility and dry deposition, as the highest OVOC deposition velocities were found for the most soluble compounds (such as HMHP and the sum of the isoprene hydroxyhydroperoxides ISOPOOH and dihydroxyepoxide IEPOX), and the lowest values among the considered species (which however did not include MVK + MACR) were found for the poorly soluble hydrogen cyanide. This finding prompted Nguyen et al. (2015) to propose a revision of Wesely's parameterization for cuticular and mesophyllic resis-

tances, which enhanced the role of Henry's law constants and appeared to match their measurements quite well. Nevertheless, the authors warned that the scheme is preliminary and that further validation is needed before it can be implemented in models with confidence. In particular, it was not tested against campaign measurements for key, but poorly soluble, compounds like MVK or MACR, PAN and ozone.





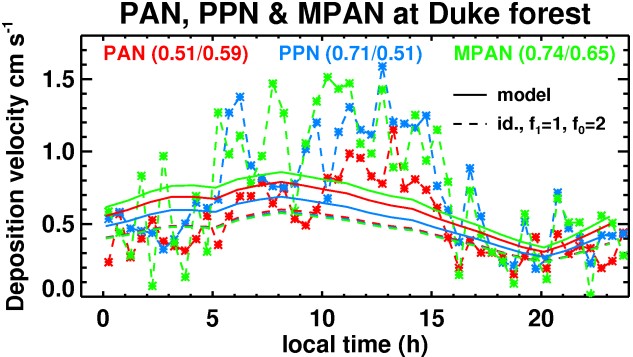

**Figure 5.** Average measured (symbols) and modelled (curves) deposition velocities (cm s$^{-1}$) of PAN, PPN and MPAN at Duke forest (see Table 5). The dash-dotted lines represent simulation results assuming $f_1 = 1$ and $f_0 = 2$ in the calculation of surface resistances. The average measured and modelled deposition velocities are given between parentheses for each compound.

In fact, there is no simple model which can reliably predict the conductance of OVOCs only as a function of their Henry's law constants and gaseous diffusivities. Whereas the deposition of hydrogen cyanide ($K_{H,298} = \sim 10$ M atm$^{-1}$ (Sander, 2015) and $D_g = \sim 0.2$ cm$^2$ s$^{-1}$ (Fuller et al., 1969)) was found to be slow, of the order of 0.3 cm s$^{-1}$ around midday at Brent (Nguyen et al., 2015), other compounds, as soluble or even less soluble than HCN, were shown to deposit much faster on vegetation. The following subsections provide an overview of reported dry deposition data for OVOCs, used for evaluation and adjustment of the deposition model.

### 3.5.1 PAN-like compounds

PAN-like compounds (e.g. PAN, PPN and MPAN) were shown to deposit fairly fast, at midday velocities of $\sim 1$ cm s$^{-1}$ (Turnipseed et al., 2006), in spite of their lower diffusivities ($D_g = \sim 0.09$ cm$^2$ s$^{-1}$) and Henry's law constants (2–4 M atm$^{-1}$) (Sander, 2015) compared to HCN. The model of Wesely (or Zhang et al. (2003)) strongly underestimates their deposition velocity at Duke forest in North Carolina, even when the reactivity factor $f_0$ is taken equal to 1 Fig. 5); the calculated values would be even much lower, by an order of magnitude, with $f_0 = 0$ and $f_1 = 1$. The field data clearly indicate the absence of any limitation to stomatal uptake, i.e. $R_m$ appears to be negligible for those compounds. Furthermore, a large fraction of their deposition is non-stomatal (Turnipseed et al., 2006; Wu et al., 2012). The model achieves a much better match with the data when assuming $f_1$ of the order of $10^4$ for PAN (also PPN, peroxypropionyl nitrate) and $5 \cdot 10^4$ for MPAN. The calculated deposition velocities remain too low around midday, likely due to overestimated stomatal resistances at that site ($\sim 800$ s m$^{-1}$ for PAN at midday in July 2007). The calculated non-stomatal component of the total conductance around midday is $\sim 0.5$ cm s$^{-1}$ for PAN, in agreement with the measurement-based estimation (Turnipseed et al., 2006). Turnipseed et al. (2006) found a significant enhancement of non-stomatal conductance under wet conditions (after a rain or when RH was very high), especially during the night. This is reflected by the observed slow increase in deposition velocity throughout the night, from $\sim 0.25$ cm s$^{-1}$ at sunset to $\sim 0.6$ cm s$^{-1}$ at sunrise in the case of PAN (Fig. 5), a feature quite well reproduced by the model. Furthermore, the model predicts a faster nighttime uptake of PAN compared to O$_3$, by almost a factor of 2, in accordance with field studies (Shepson et al., 1992; McFayden and Cape, 1999). Note that the model assumptions adopted by Paulot et al. (2018), ($f_1 = 0$ and $f_0 = 2$) lead to an slightly larger overall underestimation of the deposition velocities and in particular of the daytime non-stomatal component (by a factor of 2), but it reproduces very well the average nighttime deposition rates.



**Table 3.** Ozone dry deposition velocity measurements used in this work. LAI is the reported leaf area index (m$^2$ m$^{-2}$), when available (values in parentheses are assumed), LT is local time, $v_d^{\mathrm{obs}}$ and $v_d^{\mathrm{mod}}$ the average measured and modelled deposition velocity, respectively. FNS stands for Fazenda Nossa Senhora. References: 1, Wu et al. (2011); 2, Padro et al. (1991); 3, Fares et al. (2014); 4, Finkelstein et al. (2000); 5, Kurpius et al. (2002); 6, Kumar et al. (2002); 7, Rannik et al. (2012); 8, Hole et al. (2004); 9, Mikkelsen et al. (2000); 10, Fan et al. (1990); 11, Rummel et al. (2007); 12, Karl et al. (2010); 13, Fowler et al. (2011); 14, Cros et al. (2000); 15, Stella et al. (2011); 16, Meyers et al. (1998); 17, Wesely (1978); 18, Padro et al. (1994a); 19, Padro et al. (1994b); 20, Rummel et al. (2007) and Kirkman et al. (2002). Notes: [a], gradient method; [b], eddy covariance method.

| Site name | Dominant plant functional type | Latitude N and longitude E (°) | period | LT hour | LAI | $v_d^{\mathrm{obs}}$ ($v_d^{\mathrm{mod}}$) cm s$^{-1}$ | Ref. |
|---|---|---|---|---|---|---|---|
| Harvard forest, Mass. | broadleaf deciduous forest | 42.54, -72.2 | 06-08/2000 | 0-24 | 3.5 | 0.50 (0.47) | 1 |
| | | | 09-10/2000 | 0-24 | 3.5 | 0.59 (0.38) | 1 |
| Borden, Ontario | broadleaf deciduous forest | 44.3, -80.9 | 07-08/1991 | 0-24 | 5 | 0.58 (0.39) | 2 |
| Castelporziano, Italy | Holm oak forest | 41.7, 12.4 | 03-05/2013 | 0-24 | 4.8 | 0.34 (0.36) | 3 |
| | | | 06-08/2013 | 0-24 | 4.8 | 0.23 (0.21) | 3 |
| Kane, Pennsylvania | broadleaf deciduous forest | 41.6, -78.8 | 06-08/1997 | 0-24 | 6.5 | 0.83 (1.05) | 4 |
| Sand Flats, New York | mixed forest | 43.6, -75.2 | 06-09/1998 | 0-24 | 6.5 | 0.82 (0.94) | 4 |
| Duke forest, N. Carolina | loblolly pine plantation | 35.97, -79.1 | 04-05/1996 | 0-24 | 3 | 0.39 (0.41) | 4 |
| Blodgett, California | Ponderosa pine plantation | 38.8, -120.7 | 06-08/1999 | 0-24 | 3.6 | 0.30 (0.23) | 5 |
| Niwot Ridge, Colorado | coniferous forest | 40.1, -105.6 | 06/2002 | 0-24 | 4.2 | 0.28 (0.30) | 6 |
| Hyytiälä, Finland | coniferous forest | 61.8, 24.3 | 06-08/2002 | 0-24 | 6 | 0.42 (0.47) | 7 |
| Hurdal, Norway | coniferous forest | 60.4, 11.1 | 06-08/2000-03 | 0-24 | 4.5 | 0.33 (0.44) | 8 |
| Ulborg, Denmark | coniferous forest | 56.3, 8.4 | 06/1994 | 0-24 | (5) | 0.51[a] (0.59) | 9 |
| | | | | | | 0.78[b] (0.59) | 9 |
| Ducke, Amazonas | tropical evergreen forest | -2.6, -60.1 | 04-05/1987 | 0-24 | 7 | 0.95 (0.95) | 10 |
| Rebio Jarú, Rondônia | tropical evergreen forest | -10.1, -61.9 | 05/1999 | 0-24 | 5.6 | 0.65 (0.58) | 11 |
| | | | 09-10/1999 | 0-24 | 5.6 | 0.85 (0.69) | 11 |
| Cuieiras, Amazonas | tropical evergreen forest | -2.6, -60.1 | 02/2008 | 10-14 | 5.5 | 2.3 (1.85) | 12 |
| Bukit Atur, Borneo | tropical evergreen forest | 4.9, 117.8 | 04/2008 | 9-15 | 6 | 0.26 (1.26) | 13 |
| | | | 06-07/2008 | 9-15 | 6 | 0.94 (1.32) | 13 |
| Enyélé, Congo | tropical evergreen forest | 2.8, 18.1 | 11-12/1996 | 10-13 | 6 | 1.5 (1.47) | 14 |
| Grignon, France | maize field | 48.8, 1.9 | 05-08/2008 | 0-24 | 4 | 0.43 (0.46) | 15 |
| Lamasquère, France | maize field | 43.7, 1.4 | 06-08/2008 | 0-24 | 3.2 | 0.33 (0.41) | 15 |
| Bondville, Illinois | maize field | 40.1, -88.4 | 08/1994 | 0-24 | 3 | 0.32 (0.33) | 16 |
| Sangamon, Illinois | maize field | 39.8, -88.8 | 07/1976 | 9-18 | (3) | 0.41 (0.46) | 17 |
| Fresno, California | vineyard | 36.8, -120.1 | 07-08/1991 | 0-24 | 3 | 0.31 (0.28) | 18 |
| Fresno, California | senescent grass | 37.1, -119.8 | 07-08/1991 | 0-24 | 1 | 0.09 (0.23) | 19 |
| Sand Mountain, Alabama | grassland | 34.3, -85.97 | 04-06/1995 | 0-24 | 1.65 | 0.24 (0.41) | 4 |
| FNS, Rondônia | pasture | -10.7, -62.3 | 05/1999 | 0-24 | 2.1 | 0.39 (0.35) | 20 |
| | | | 10/1999 | 0-24 | 1.2 | 0.35 (0.27) | 20 |



**Table 4.** Sulfur dioxide dry deposition velocity measurements used in this work. LAI is the reported leaf area index (m$^2$ m$^{-2}$), when available (values in parentheses are assumed), LT is the local time (hour), $v_d^{\mathrm{obs}}$ and $v_d^{\mathrm{mod}}$ the average measured and modelled deposition velocity, respectively. Notes: [a]: $v_d^{\mathrm{obs}}$ estimated from branch enclosure measurements (see text). References: 1, Padro et al. (1993); 2, Finkelstein et al. (2000); 3, Fowler and Unsworth (1979); 4, Meyers et al. (1998); 5, Hicks et al. (1986).

| Site name | Dominant plant functional type | Latitude N and longitude E (°) | period | LT hour | LAI | $v_d^{\mathrm{obs}}$ ($v_d^{\mathrm{mod}}$) cm s$^{-1}$ | Ref. |
|---|---|---|---|---|---|---|---|
| Borden, Ontario | broadleaf deciduous forest | 44.3, -80.9 | 03-04/1990 | 8-17 | (0.3) | 0.5 (0.53) | 1 |
| | | | | 21-6 | (0.3) | 0.1 (0.36) | 1 |
| Kane, Pennsylvania | broadleaf deciduous forest | 41.6, -78.8 | 06-08/1997 | 9-15 | 6.5 | 1.09 (1.06) | 2 |
| | | | | 20-4 | 6.5 | 0.25 (0.27) | 2 |
| Sand Flats, New York | mixed forest | 43.6, -75.2 | 06-09/1998 | 9-15 | 6.5 | 1.1 (1.07) | 2 |
| | | | | 20-4 | 6.5 | 0.17 (0.31) | 2 |
| Nottingham, UK | wheat crop | 52.8, -1.1 | 05-07/1974 | 8-18 | 4.5 | 1.0 (0.99) | 3 |
| | | | | 18-8 | 4.5 | 0.64 (0.84) | 3 |
| Bondville, Illinois | maize field | 40.1, -88.4 | 08/1994 | 0-24 | 3 | 0.66 (0.44) | 4 |
| Sand Mountain, Alabama | grassland | 34.3, -85.97 | 04-06/1995 | 0-24 | 1.65 | 0.58 (0.49) | 4 |
| Southern Ohio | grassland | 39.8, -83.6 | 09/1979 | 11-16 | (3) | 0.65 (0.68) | 5 |





**Table 5.** Dry deposition velocity data of OVOCs used in this study. LAI is the reported leaf area index (m$^2$ m$^{-1}$), when available (values in parentheses are assumed), LT is the local time, $v_d^{\mathrm{obs}}$ and $v_d^{\mathrm{mod}}$ are the average measured and modelled deposition velocity. When the measurement refers to a sum of several compounds, the model value is an average weighted by their model-calculated relative abundance. MVKMAC stands for the sum MVK+MACR+0.44 ISOPOOH; MVKN, for MVKNO3+MACRNO3. References: 1, Schade et al. (2011); 2, Karl et al. (2005); 3, Rantala et al. (2015); 4, Laffineur et al. (2012); 5, Schallhart et al. (2016); 6, Karl et al. (2004); 7, Langford et al. (2010); 8, Misztal et al. (2011); 9, (Valverde-Canossa et al., 2006); 10, Hall and Claibron (1997); 11, Karl (2009b); 12, Karl et al. (2010); 13, Andreae et al. (2002); 14, Turnipseed et al. (2006); 15, Wolfe et al. (2009); 16, Rottenberger et al. (2004); 17, Kuhn et al. (2002); 18, Sanhueza et al. (1992); 19, Nguyen et al. (2015); 20, Russo et al. (2010). Notes: $^a$, corrected for non-deposition fluxes (see text); $^b$, estimated from branch enclosure measurements, assuming $R_a + R_b = 40$ s m$^{-1}$.

| Site | Dominant plant functional type | Latitude N and longitude E (°) | month(s)/ year(s) | LT hour | LAI | Species | $v_d^{\mathrm{obs}}$ ($v_d^{\mathrm{mod}}$) cm s$^{-1}$ | Ref. |
|---|---|---|---|---|---|---|---|---|
| Soroe, Denmark | beech forest | 55.4, 11.7 | 6/2007 | 0-24 | 5 | CH$_3$OH | 1.1 (1.07) | 1 |
| Duke forest, NC | pine plantation | 35.98, -79.1 | 7/2003 | 22-5 | 3 | CH$_3$OH | 0.96 (0.80) | 2 |
| Hyytiälä, Finland | coniferous forest | 61.8, 24.3 | 5/2010-13 | 0-24 | 6 | CH$_3$OH | 0.18 (0.44) | 3 |
| | | | 9/2010-13 | 0-24 | 6 | CH$_3$OH | 0.50 (0.46) | 3 |
| Vielsalm, Belgium | mixed forest | 50.3, 5.99 | 7-9/2009 | 0-24 | (3.5) | CH$_3$OH | 1.78 (1.25) | 4 |
| Bosco Fontana, It. | oak forest | 45.2, 10.74 | 6-7/2012 | 2-8 | (3.5) | CH$_3$OH | 0.21 (0.3) | 5 |
| Costa Rica | tropical rainforest | 10.4, -83.9 | 4-5/2003 | 18-6 | 4.2 | CH$_3$OH | 2.9 (1.09) | 6,2 |
| Bukit Atur, Borneo | tropical rainforest | 4.9, 117.8 | 4-5/2008 | 0-24 | 6 | CH$_3$OH | 0.36 (0.62) | 7 |
| | | | 6-7/2008 | 0-24 | 6 | CH$_3$OH | 0.57 (0.63) | 7 |
| Borneo | oil palm plantation | 5.2, 118.4 | 5-6/2008 | 0-24 | 6 | CH$_3$OH | 1.25 (1.38) | 8 |
| | | | | 10-14 | | CH$_3$OH | 2.3 (2.08) | 8 |
| Bavaria, Germany | spruce forest | 50.0, 11.7 | 7/2001 | 0-24 | (5) | CH$_3$OOH | ∼1.2 (1.11) | 9 |
| Boardman, Sask. | jack pine forest | 53.9, -104.7 | 6-8/1994 | 9-18 | 2.1 | ROOH | 1.6 (1.84) | 10 |
| Duke forest, NC | pine plantation | 35.98, -79.1 | 7/2003 | 10-17 | 3 | MVKMAC | 0.33 (1.89) | 2 |
| | | | | 22-5 | | MVKMAC | 2.4 (1.39) | 2 |
| Borneo | oil palm plantation | 5.2, 118.4 | 5-6/2008 | 8-17 | 6 | MVKMAC | 0.87 (2.17) | 8 |
| Costa Rica | tropical rainforest | 10.4, -83.9 | 4-5/2003 | 10-14 | 4.2 | MVKMAC | 1.6 (3.28) | 6,11 |
| Cuieiras, Amazonas | tropical rainforest | -2.6, -60.2 | 2/2008 | 10-14 | 5.5 | MVKMAC | 2.4 (2.09) | 8,12 |
| Cuieiras, Amazonas | tropical rainforest | -2.6, -60.1 | 11/1999 | 10-14 | 5.5 | MVKMAC | 2.4 (1.68) | 13 |
| Duke forest, NC | pine plantation | 35.98, -79.1 | 7/2003 | 0-24 | 3 | PAN | 0.51 (0.59) | 14 |
| | | | | | | MPAN | 0.74 (0.65) | 14 |
| | | | | | | PPN | 0.71 (0.51) | 14 |
| Blodgett, CA | pine plantation | 38.8, -120.7 | 9/2007 | 10-14 | 5.1 | PAN | 0.3-0.5$^a$ (0.32) | 15 |
| | | | | | | MPAN | 0.3-0.5$^a$ (0.29) | 15 |
| | | | | | | PPN | 0.5$^a$ (0.34) | 15 |





| Site | Dominant plant functional type | Latitude N and longitude E (°) | month(s)/ year(s) | LT hour | LAI | Species | $v_d^{\mathrm{obs}}$ ($v_d^{\mathrm{mod}}$) cm s$^{-1}$ | Ref. |
|------|-------------------------------|-------------------------------|-------------------|---------|-----|---------|-----------------------------------------------------------|------|
| Jarú, Rondônia | tropical rainforest | -10.1, -62.9 | 5,9-10/1999 | 8-17 | 5.5 | CH$_2$O | 0.71 (0.69) | 16[b] |
| | | | | | | CH$_3$CHO | 1.00 (1.10) | 16[b] |
| Cuieiras, Amazonas | tropical rainforest | -2.6, -60.2 | 2/2008 | 10-14 | 5.5 | CH$_3$CHO | 1.40 (1.80) | 11,12 |
| Duke forest, NC | pine plantation | 35.98, -79.1 | 7/2003 | 22-5 | 3 | CH$_3$CHO | 1.29 (1.02) | 2 |
| | | | | 10-17 | 3 | CH$_3$CHO | 2.3 (1.12) | 2 |
| Duke forest, NC | pine plantation | 35.98, -79.1 | 7/2003 | 22-5 | 3 | CH$_3$COCH$_3$ | 0.46 (0.45) | 2 |
| Cuieiras, Amazonas | tropical rainforest | -2.6, -60.2 | 2/2008 | 10-14 | 5.5 | HYAC | 2.1 (1.99) | 11,12 |
| | | | | | | GLYALD | 2.1 (2.13) | 11,12 |
| Jarú, Rondônia | tropical rainforest | -10.1, -62.9 | 5,9-10/1999 | 8-17 | 5.5 | HCOOH | 0.83 (1.09) | 17 |
| | | | | | | CH$_3$COOH | 0.74 (0.88) | 17 |
| Duke forest, NC | pine plantation | 35.98, -79.1 | 7/2003 | 22-5 | 3 | CH$_3$COOH | 1.3 (0.62) | 2 |
| | | | | 10-17 | 3 | CH$_3$COOH | 3.1 (0.57) | 2 |
| Bukit Atur, Borneo | tropical rainforest | 4.9, 117.8 | 4-5/2008 | 0-24 | 6 | CH$_3$COOH | 0.14 (0.52) | 6 |
| Venezuela | cloud forest | 10.5, -66.9 | 3-12/1989 | 21-6 | 4 | CH$_3$COOH | 0.68 (1.21) | 18 |
| | | | | | | HCOOH | 1.1 (1.98) | 18 |
| Brent, AL | deciduous forest | 32.9, -87.24 | 06/2013 | 0-24 | 4.7 | HCOOH | 0.43 (0.44) | 19 |
| | | | | | | HMHP | 1.64 (1.55) | 19 |
| | | | | | | IEPOX + ISOPOOH | 1.04 (1.19) | 19 |
| | | | | | | PAA | 1.11 (1.16) | 19 |
| | | | | | | HPALD | 1.01 (1.12) | 19 |
| | | | | | | ISOPN | 0.64 (0.71) | 19 |
| | | | | | | MVKN | 0.59 (0.63) | 19 |
| | | | | | | NOA | 0.75 (0.72) | 19 |
| | | | | | | HYAC | 0.74 (0.66) | 19 |
| | | | | | | BIACETOH | 0.46 (0.39) | 19 |
| | | | | | | DHBO | 0.47 (0.44) | 19 |
| | | | | | | NISOPOOH | 0.58 (0.55) | 19 |
| | | | | | | APINONO2 | 0.30 (0.39) | 19 |
| New Hampshire | mixed forest | 43.1, -70.95 | 6-8/2002 | 21-6 | 3.5 | CH$_3$ONO$_2$ | 0.13 (0.10) | 20 |

The observations clearly suggest that liquid water in leaves or needles accelerates a form of reactive uptake of PAN-like compounds by vegetation. The higher $f_1$ value for MPAN might reflect a higher reactivity in leaf/needle water due to the presence of the double bond in the molecule. Caution is warranted, however, since PPN was also found to deposit faster than PAN. The measurements might be affected by chemical production and loss within the canopy (Farmer and Cohen, 2008). Although the chemical contributions to the observed fluxes were estimated to be small at Duke forest by Turnipseed et al. (2006), they were found to be significant in a field study at a pine plantation



(Blodgett forest) in California (Wolfe et al., 2009) (Table 5). On one hand, the thermal loss gradient was estimated to contribute substantially to the measured exchange velocity, defined as the ratio of the eddy covariance flux to the concentration. This contribution is small at night but reaches $\sim 0.2$ cm s$^{-1}$ around midday, and even much more during warm periods (Wolfe et al., 2009). On the other hand, a larger photochemical production of PAN and MPAN compared to PPN was proposed to explain the substantial difference between the measured

exchange velocities of PAN or MPAN (0.3-0.4 cm s$^{-1}$) and PPN (0.7 cm s$^{-1}$). A even larger difference (factor of three) was derived when considering only the warm measurement period. The observed exchange velocity of PPN, i.e. $\sim 0.5$ cm s$^{-1}$ after correction for the thermal decomposition gradient might be a more realistic estimation of the deposition velocity of PAN-like compounds at this site. The conclusion of a strong role of photochemical production of peroxy nitrates at Blodgett forest is supported by the large observed upward fluxes of peroxy nitrates ($\Sigma$ PNs) at this site in August 2005 (Farmer and Cohen, 2008) which indicated a strong photochemical production favoured by high

within-canopy OH levels ($\sim 3 \cdot 10^7$ molec. cm$^{-3}$) and high emissions of very reactive BVOC compounds. The photochemical production of PAN and MPAN could possibly also explain the faster deposition of PPN compared to PAN at Duke forest, if the OH levels in the canopy were high enough to cause a strong gradient in the photochemical production of PAN and MPAN.

To summarize, our model adoption of high $f_1$ values for PAN-like compounds appears justified as it leads to a fair agreement at Duke forest (acknowledging the stomatal conductance underestimation) and to a moderate underestimation at Blodgett forest (Table 5). The latter

is partly due to an underestimation of the stomatal conductance (0.08 vs. 0.12 cm s$^{-1}$ for PAN around midday). A fairly good performance of the model could also be achieved by adopting a lower $f_1$ with $f_0 > 1$ as in Paulot et al. (2018). More measurements will be needed to further validate and refine the parameterization.

### 3.5.2 MVK+MACR

The major isoprene oxidation products, MVK and MACR, are only slightly more soluble than PAN. Their uptake by leaves of different

*Quercus* species was shown to be a significant sink in fumigation experiments by Tani et al. (2010), which indicated very low values of the ratio of intercellular to external concentration for MACR (also crotonaldehyde) and to a lesser extent for MVK. The fast deposition of MVK + MACR was also indicated by several field studies at tropical and mid-latitude forest sites, with daytime deposition velocities ranging between $\sim 1$ and 2.5 cm s$^{-1}$ (Table 5 and Fig. 3), well matched by the model when adopting $f_1 = 5 \cdot 10^4$, i.e. the same value as for MPAN. Caution is needed due to the potential interference of isoprene hydroxyhydroperoxides (ISOPOOH) in the measurement of MVK + MACR

(Liu et al., 2013; Rivera-Rios et al., 2014; Bernhammer et al., 2017). The precise dependence of this effect on the instrument configuration is complex and unfortunately not well established for the flux measurement campaigns used in this study. Here we assume a conversion efficiency of 44% as reported for the 1,2-ISOPOOH isomer (Rivera-Rios et al., 2014) and consistent with the upper limit of 50% derived by Wolfe et al. (2016) for ISOPOOH. The precise value of this parameter has only a minor influence on the model results due to the lower reactivity and generally higher abundance of MVK + MACR compared to ISOPOOH. In particular, the high deposition velocities measured

over Amazonia, Borneo, Costa Rica and North Carolina (Table 5) cannot be explained by the interference alone, based on the parameterized deposition velocities of ISOPOOH constrained by recent field measurements in Alabama (see further below) and on the global model results indicating a maximum contribution of ISOPOOH to the observed signal of about 20% at the measurement sites (assuming 44% conversion). The model values shown in Table 5 and Fig. 3 are average deposition velocities for the sum MVK+MACR+0.44 ISOPOOH, weighted by the model-calculated concentrations.

Another complicating issue is the evidence of MVK and MACR emission which could outbalance or at least mask part of the MVK + MACR deposition flux. Although this emission was found to be negligible for several broadleaved deciduous tress (Fares et al., 2015), (small but) positive fluxes were found at a tropical forest in Borneo (Langford et al., 2010), at deciduous forests in Europe (Spirig et al.,

...





2005; Kalogridis et al., 2014; Brilli et al., 2016; Schallhart et al., 2016) and at an orange orchard in California (Park et al., 2013). Emission of MVK or MACR might be caused by the oxidation of isoprene in leaves (Jardine et al., 2012; Fares et al., 2012; Cappellin et al., 2017), by the oxidation of diterpenoids emitted by plant trichomes (Jud et al., 2016) and/or by the decomposition of isoprene hydroxyhydroperoxides on plant surfaces, similar to their reaction on the metal surfaces of analytical instruments (Bernhammer et al., 2017; Misztal et al., 2016).

Therefore, daytime deposition velocities based on flux measurements (Table 5) should be considered as lower limits. This might possibly explain some of the model overestimation of deposition velocities during daytime (Duke forest, Costa Rica), although this remains speculative. At the oil palm plantation in Borneo, the model overestimation for MVK + MACR is at least partly in line with the overestimation for ozone (Fig. 3(c)) which suggests a large model underestimation of both stomatal and cuticular resistances for that ecosystem. Regardless of the model failure, the much higher deposition velocity observed for MVK+MACR compared to ozone at that site corroborates the existence of a

large non-stomatal component to their deposition, justifying the adoption of a high $f_1$ value for those compounds. Indeed, their low solubility and lower diffusivity compared to ozone (0.095 vs. 0.18 cm$^2$ s$^{-1}$) would lead to a lower deposition velocity compared to ozone if $f_1$ was of the order of 1. At Cuieiras in Amazonia during the AMAZE campaign (Karl, 2009a, b; Karl et al., 2010), the deposition velocities of O$_3$ and MVK+MACR were very similar and both very high ($\sim$2 cm s$^{-1}$). However, whereas the ozone deposition flux was primarily stomatal, the flux of MVK+MACR had a strong non-stomatal component, as seen from the large underestimation of the deposition velocity calculated

with $f_0 = f_1 = 1$ (Fig. 3(d)). The same holds for the deposition velocity of hydroxyacetone (HYAC), which would also be underestimated by Wesely's model with $f_0 = f_1 = 1$, in spite of its higher Henry's law constant. Note that the moderate model underestimation of the ozone deposition velocity (1.85 vs. 2.3 cm $^{-1}$) could have a number of reasons, including the gas-phase reaction of ozone with reactive terpenoids (as discussed in Sect. 3.4) or with nitric oxide emitted by the soil, or the surface reaction of ozone with semi-volatile diterpenoid compounds (Jud et al., 2016).

Note that the high $f_1$ value implies very low mesophyllic resistances ($R_m << 1$ s m$^{-1}$), whereas small but non-negligible limitation to stomatal uptake found by Tani et al. (2010) for oak saplings, especially for MVK. Further work is needed to further elucidate the processes and constrain the parameters of MVK and MACR deposition.

### 3.5.3 Acetaldehyde, acetone and formaldehyde

Acetaldehyde is another poorly soluble organic compound for which fast deposition (1–2.3 cm s$^{-1}$) was observed at several forest sites

(Table 5). Acetaldehyde fluxes are generally bidirectional, i.e. it is also emitted by foliage (see Millet et al. (2010) and references therein), implying the existence of a compensation point, defined as the ambient concentration above which there is net deposition, and below which emission dominates. The compensation point ranged between $\sim$0.4 and $> 1$ ppbv above a tropical forest (Rottenberger et al., 2004) as well as above ryegrass (Custer and Schade, 2007), but it takes much higher values (6 ppb) e.g. over spruce under warm conditions (6 ppbv) (Cojocariu et al., 2004). Rottenberger et al. inferred a canopy resistance of $\sim$50-70 s m$^{-1}$ at a tropical rainforest site (Jarú), based on the

observed dependence of the fluxes on ambient concentrations. Furthermore, whereas stomatal conductance dominated the total exchange during the wet season (May), a substantial part of the flux (up to $\sim$50%) was thought to be non-stomatal during the dry season under stress conditions, when the stomata were largely closed due to high temperatures and the resulting large vapour pressure deficit. The observations are fairly well reproduced by adopting $f_1 = 2 \cdot 10^4$ for CH$_3$CHO. The $v_d$ overestimation at Cuieiras (1.8 vs. 1.4 cm $^{-1}$) is not unexpected since the reported exchange velocity is a net flux including a potentially significant emission component. The high $f_1$ value for acetalde-

hyde is qualitatively consistent with its faster-than-expected uptake by cloud droplets suggesting the formation of a surface complex (aldol condensation) at the water-air interface (Jayne et al., 1992).





There is unfortunately little data on the deposition of acetone and formaldehyde, to a large degree because biogenic emission frequently dominates over deposition. To our knowledge, daytime acetone deposition has not been documented. At Duke forest, weak daytime deposition in the lower canopy was more than compensated by upper-canopy emissions favoured by high visible radiation levels (Karl et al., 2005). At night, significant deposition occurred, although its estimated exchange velocity ($\sim$0.46 cm s$^{-1}$) was about twice lower than that of methanol,

presumably because of its lower solubility in water.

In spite of the much higher solubility of formaldehyde ($K_H^{298}$ = 3200 M atm$^{-1}$) compared to acetaldehyde and acetone, its deposition was found to be slow, and it appears to proceed primarily through the stomata, i.e. cuticular resistance is likely high (Rottenberger et al., 2004; Seco et al., 2008; Brilli et al., 2014). The formaldehyde fluxes are generally bidirectional, with primary emissions lower than those of acetaldehyde or acetone for Norway spruce (Cojocariu et al., 2004; Müller et al., 2006), but dominant among small carbonyls for Eucalyptus

(Winters et al., 2009). The mesophyll resistance ($R_m$) is believed to be non-negligible, as it was found to be of the same order as the stomatal resistance at Jarú (Rottenberger et al., 2004). The limited available data therefore suggest a low value for $f_1$ ($\sim$0.03), and $f_0 = 0$.

### 3.5.4   Methyl hydroperoxide

There is little data available on the deposition of methyl hydroperoxide. The large scatter and high uncertainties of the flux measurements at a spruce forest in Bavaria (Valverde-Canossa et al., 2006) preclude the derivation of a reliable estimate. Nevertheless, the ratio of the average

reported flux (0.03$\pm$0.03 nmol m$^{-1}$ s$^{-1}$) to the mean concentration ($\sim$0.07 ppb) implies a deposition velocity close to $\sim$1 cm s$^{-1}$, fairly high considering the moderate Henry's law constant of $CH_3OOH$ (310 M atm$^{-1}$), but much lower than the average measured deposition velocity of $H_2O_2$ (5 cm s$^{-1}$) for which surface resistance is negligible (Valverde-Canossa et al., 2006). Hall and Claibron (1997) measured the deposition of the total organic peroxides (ROOH), among which $CH_3OOH$ is a major component at the measurement site according to our model calculations (53%) as well as according to speciated organic hydroperoxide measurements at the site. Besides a small contribution

of higher alkanoic hydroperoxides (3%), the rest consists mostly in more soluble compounds such as peracids and hydroxy hydroperoxides for which deposition to vegetation was recently shown to be very fast (Nguyen et al., 2015) (Fig. 6). Given these constraints on the deposition of functionalized hydroperoxides, the ROOH and $CH_3OOH$ deposition data are reasonably well reproduced (Table 5) by the model when adopting $f_1$ = 400 for non-functionalized hydroperoxides. Note that at both sites (Bavaria and Saskatchewan), the model captures well the measured high deposition velocity of $H_2O_2$ (around 5 cm s$^{-1}$ during the day), when adopting a very high $f_1$ value ($10^4$).

### 3.5.5   Methanol

The exchanges of methanol are bidirectional, with a large emission component (especially during daytime) dependent on leaf age and usually highest around midday (Stavrakou et al., 2011; Laffineur et al., 2012; Wohlfahrt et al., 2015), although nighttime emissions were also reported (Schade et al., 2011). At some sites, however, deposition may dominate over emission (Laffineur et al., 2012; Misztal et al., 2011). Methanol deposition is strongly enhanced in humid conditions, indicating that dew and/or needle/leaf/soil water plays an important

role. Methanol may be adsorbed or dissolved in this water, and eventually be degraded and removed (Karl et al., 2004). Methanol taken up by dew is potentially released back to the gas-phase upon evaporation, however. Deposition velocities often exceeding 1 cm s$^{-1}$ were reported from the analysis of methanol flux measurements at forest sites in tropical regions and at mid-latitudes (Table 5). The highest values at mid-latitudes were found at a mixed forest site near Vielsalm in Belgium (1.78 cm s$^{-1}$ on average, 2.4 cm s$^{-1}$ in wet conditions), consistent with a large, strongly humidity-dependent non-stomatal component. In contrast, much lower deposition velocities ($\sim$0.2 cm s$^{-1}$) are derived from

the reported early morning fluxes at an oak forest (Bosco Fontana) in Italy, assuming emissions to be still small at these hours (2-8 LT). The low relative humidities and friction velocities at Bosco Fontana might explain the large difference relative to Vielsalm, given the influence of





**Figure 6.** Average measured and modelled deposition velocities at Brent, Alabama. Compound notation as in Table 1 except ISOPN = ISOPBNO3 + ISOPDNO3, ISOPOOH = ISOPBOOH + ISOPDOOH, NISOPOOH = NISOPOOHD + NISOPOOHB, HPALD = HPALD1 + HPALD2. When the data refer to the sum of several compounds, the model value is an average weighted by their model-calculated relative abundances. The dotted lines represent the calculated values for individual compounds. The dash-dotted lines represent results assuming $f_0 = f_1 = 1$ in the resistance calculation. The 24-hour averages are given in the top right corner of each plot (measurements in black, model in red, the simulation assuming $f_0 = f_1 = 1$ given in parentheses). The aerodynamic resistance used in the simulation is inferred from the measured nitric acid deposition velocity, assuming $R_c = 0$ and with $R_b$ obtained from the model.





these parameters on cuticular resistance (Sect. 3.3.3). This difference as well as the other reported methanol deposition velocities are fairly well captured by the model, when using $f_1 = 600$. At the palm plantation, the magnitude and the diurnal cycle of the deposition velocity with a marked midday maximum ($>2$ cm s$^{-1}$) are well reproduced by the model, but possibly for wrong reasons, since the modelled diurnal cycle is driven by stomatal exchange, whereas the ozone deposition data suggested very low stomatal conductances (Fig. 3). But if indeed the stomatal resistances are very high at this site, the reasons for the observed midday maximum of the methanol dry deposition velocity are unclear.

### 3.5.6  Formic and acetic acid

The effective Henry's law constant of carboxylic acids ($K_H^*$) is substantially enhanced by their dissociation in water,

$$K_H^* = K_H \cdot (1 + K_A/[\text{H}^+]) \tag{47}$$

where $K_A$ is the dissociation constant equal to $1.8 \cdot 10^{-4}$ for HCOOH and $1.7 \cdot 10^{-5}$ for CH$_3$COOH (Lide, 2000). In pure neutral water, there is therefore more than an order of magnitude difference between the effective solubility of formic ($1.6 \cdot 10^7$ M atm$^{-1}$) and acetic acid ($7 \cdot 10^5$ M atm$^{-1}$). However, the $K_H^*$ values estimated from simultaneous measurements of gas- and aqueous-phase concentrations in fog or dew at several sites (see Khare et al. (1999) and references therein) show little dependence on pH and are much more similar between the two compounds ($\sim 3 \cdot 10^4$ and $\sim 8 \cdot 10^4$ M atm$^{-1}$). The causes for these departures from theoretical equilibrium are not well understood, although hypotheses have been formulated (e.g. the presence of organic films limiting mass transport). The ratio of field-based $K_H^*$ values for the two compounds is ca. 2.5, similar to the ratio of their simple Henry's law constant $K_H$ (Sander, 2015). This justifies the use of the simple ($K_H$) instead of the effective Henry's law constants in the parameterization of the canopy resistance (Sect. 3.3.4 and Table 1).

Furthermore, this choice leads to a good model agreement with experimental estimates in the rare field studies where HCOOH and CH$_3$COOH deposition velocities were simultaneously determined. Caution is warranted, since carboxylic acid fluxes are bidirectional, with compensation points ranging between 0.16 and $\sim 2.1$ ppbv, and since vegetation might be a larger source of formic compared to acetic acid (Jardine et al., 2011b). Nevertheless, as seen in Table 5, the adoption of a unique value for $f_1$ (20) for both species leads to a good model simulation of the small difference (factor $\sim 1.1$) between their daytime deposition velocities at Jarú, Rondonia (Kuhn et al., 2002), as well as the larger $v_d$ difference at night in the Venezuelan cloud forest (factor 1.6) (Sanhueza et al., 1992). Note that biogenic emissions are usually very low at night, and that the presence of a compensation point was accounted for by Kuhn et al. (2002) in their derivation of deposition velocities at Jarú. The different $v_d$ ratios at the two campaigns are explained by the much higher humidity and friction velocity at night in the cloud forest, giving a major role to cuticular conductance (which is proportional to $K_H$) at this site, whereas deposition was primarily stomatal during daytime at Jarú (Kuhn et al., 2002), implying a stronger role for gas-phase diffusivity $D_g$. $D_g$ is only about 1.2 times larger for formic than for acetic acid, whereas their $K_H$ differs by more than a factor of 2.

Both stomatal and non-stomatal pathways were significant during daytime at Brent, Alabama. Deposition through the stomata was however dominant, as shown by the timing of the $v_d$ peak (around 9:00 LT) likely resulting from the dependence of stomatal resistance on solar radiation and the vapour pressure deficit (Sect. 3.3.2).

### 3.5.7  Higher hydroperoxides and peroxyacetic acid

As seen on Fig. 6, among the organic compounds investigated by Nguyen et al. (2015), the highest midday deposition velocities (4–5 cm s$^{-1}$) were measured for HOCH$_2$OOH (HMHP) which is at the same time among the most soluble ($K_H = \sim 2 \cdot 10^6$ M atm$^{-1}$) and fast-diffusing ($D_g = 0.12$ cm$^2$ s$^{-1}$) organic species considered. A unique value of $f_1$ (50) is assumed for hydroxyhydroperoxides (including HMHP and





ISOPOOH) and for the functionalized epoxides (including IEPOX). The estimation is crude deposition velocities are only weakly dependent on $f_1$ at high $f_1$ and because of the large uncertainties in the Henry's Law constants; for example, our $K_H$ for IEPOX and ISOPOOH are about 30 times lower than the estimate by Marais et al. (2012). Our value of $\alpha$ (as defined in Eq. 45-46) for ISOPOOH ($\sim$25) is similar to the value (20) adopted by Paulot et al. (2018). For HMHP, although the simulation with $f_1 = 1$ clearly underestimates the deposition velocity

(factor of 2 around noon), the precise value of $f_1$ is unimportant as long as $R_{\text{cut}} << R_a + R_b$ ($\sim$20 s m$^{-1}$), such that the overall canopy resistance is negligible. Cuticular deposition is largely dominant for HMHP and more generally for all fast-depositing compounds, because both the stomatal resistance $R_s$ and the in-canopy transfer resistance $R_{\text{ac}}$ are much larger than $R_{\text{cut}}$.

The surface resistance for IEPOX is also expected to be negligible due to its very high estimated solubility. Its deposition is only slightly slower compared to HMHP (by 18%), due to a lower diffusivity (by 60%) resulting in a higher quasi-laminar layer resistance $R_b$. The

isoprene hydroxyhydroperoxides ISOPOOH, being somewhat less soluble, have a non-negligible cuticular resistance resulting in a lower $v_d$ (by $\sim$ 20%) compared to IEPOX. Cuticular resistances for $CH_3CO(OOH)$ (PAA) and for the hydroperoxy enals (HPALD) should also be small but non-negligible ($\sim$10-50 s m$^{-1}$) in order to explain their high observed midday $v_d$ of $\sim$3 cm $^{-1}$. The corresponding values for $f_1$ are taken equal to $5 \cdot 10^3$ and 30, respectively.

### 3.5.8 Organic nitrates

The only field-based deposition velocity estimate for simple alkyl nitrates is an average nighttime $v_d$ estimation for methylnitrate ($CH_3ONO_2$), 0.13 cm s$^{-1}$ during summer in a New Hampshire forest (Russo et al., 2010). Interestingly, the $f_1$ value (1000) required to reproduce this high value (considering its very low $K_H$ of only 2 M atm$^{-1}$) is the same as the $f_1$ value needed for the isoprene hydroxynitrate family (ISOPN on Fig. 6) and for nitroxyacetone (NOA) in comparisons with the field data at Brent (Nguyen et al., 2015) (Fig. 6). In contrast, the hydroxycarbonyl nitrates MVKNO3+MACRNO3 (including $CH_3COCH(ONO_2)CH_2OH$ and $OCHC(CH_3)(ONO_2)CH_2OH$) were

found to deposit more slowly ($<$ 2 cm s$^{-1}$) than NOA in spite of their much higher predicted solubility ($\sim 10^5$ M atm$^{-1}$ for the major isomer, $CH_3COCH(ONO_2)CH_2OH$, vs. $10^3$ M atm$^{-1}$ for NOA). This relatively slow deposition is reproduced by the model by setting $f_1 = 6$ for the trifunctional nitrates. Although the additional hydroxy group in MVKNO3+MACRNO3 would appear to justify a higher $K_H$, a substantial overestimation cannot be excluded given the large $K_H$ uncertainties for polyfunctional compounds and the noted tendency of group-estimation methods to underestimate the vapour pressure of highly oxygenated species. The $f_1$ value for such compounds will be ad-

justed when experimental $K_H$ estimations will become available. Interestingly, the value of $\alpha = f_1 \cdot K_H / 10^5$ is relatively similar (between 6 and 12) for ISOPBN, NOA and MVKNO3 (by far the major component of the sum MVKNO3 + MACRNO3), consistent with the evaluation by Paulot et al. (2018) also based on the Brent dataset.

### 3.5.9 Other compounds

As in the case of organic nitrates, the simpler hydroxycarbonyls (HYAC and GLYALD) are found to deposit faster (maximum values of

$\sim$2 cm s$^{-1}$ in Cuieiras and in Brent) than the trifunctional compounds for which data is available (BIACETOH, $CH_3COCOCH_2OH$ and DHBO, $CH_3COCH(OH)CH_2OH$) (Fig. 6 and Table 5), in spite of the much higher estimated $K_H$ of the latter compounds. As above, this might reflect a large $K_H$ overestimation for such compounds, although a larger emission, partially masking the deposition flux, could also explain the difference, or the simpler (difunctional) compounds might be more efficiently consumed by the plants.

Regarding the other compounds, the $f_1$ values used in the model (Table 1) are extrapolated from similar compounds for which experimental

data is available. In particular, given the general pattern of low $f_1$ values (of the order of 1) required to match experimental deposition data for trifunctional compounds (except IEPOX), we adopt $f_1 = 1$ for all compounds with three or more functionalities, not counting the epoxide

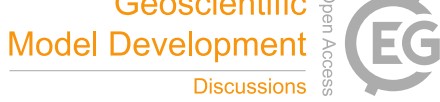

and lactone groups. $f_1 = 1$ is also assumed for the simple dicarbonyls (glyoxal and methylglyoxal), for which the field-based nighttime dry deposition velocity estimate by Huisman et al. (2011) (0.3 cm s$^{-1}$ for glyoxal at Blodgett forest in September 2007) remains overestimated by the model (0.5 cm s$^{-1}$). This might be due to an underestimation of the combined aerodynamic/quasi-laminar boundary layer resistance (also suggested by the comparison for ozone at that site, see Fig. 2(d)), or to uncertainties in the field-based estimate which is based on

the observed nighttime decay of near-surface glyoxal concentration. Clearly, more flux measurements are needed for a large number of oxygenated VOCs for which little or no data is currently available.

## 4 Regional and global modelling

### 4.1 Model description and simulations

The MAGRITTE model calculates the distribution of 175 chemical compounds either globally at 2° (latitude) × 2.5° (longitude) resolution,

or regionally at 0.5°×0.5° resolution. The meteorological fields are provided by ECMWF ERA-Interim analyses (Dee et al., 2011). Most model parameterizations, including the transport scheme, inherit from the IMAGES model (Stavrakou et al., 2009a, b, 2015; Bauwens et al., 2016). The trace gas emissions and the chemical degradation scheme of biogenic VOCs are presented in detail in a companion paper (Müller et al., 2018). Oceanic emissions of methanol, acetaldehyde and acetone are calculated as described in Sect. 3.3.1.

As in Müller et al. (2018), MAGRITTE is run for a period of 18 months starting in July 2012 at the global scale (2°×2.5° resolution) and

at the regional scale for the U.S. (0.5° resolution). Only the results for the year 2013 are discussed here. Two simulations are conducted. In the standard run (STD), all NMVOC sources are considered, with dry deposition velocities calculated as described in the previous sections. In order to separate the contributions of NMVOCs and methane oxidation to the dry deposition fluxes, an additional run is conducted (METHONLY) neglecting all NMVOC sources. The latter simulation uses oxidant fields (OH, HO$_2$, NO, NO$_2$, NO$_3$ and O$_3$ concentrations) calculated by the STD run, so that the difference between the two simulations (STD - METHONLY) represents the impact of non-methane

VOCs.

### 4.2 Model results

Figure 7 illustrates the annually averaged deposition velocity distribution calculated for a few compounds at the global scale. Over oceans, the highest values (∼1.5 cm s$^{-1}$) are found for fast-diffusing, soluble compounds (e.g. CH$_3$OH and HCOOH) at around 50-60° latitudes of both hemispheres, which are characterized by the highest winds. Unsurprisingly, over continents as well, the highest deposition velocities

are found for the most soluble compounds, HMHP and IEPOX (2–5 cm s$^{-1}$ over forests). However, even very poorly soluble compounds like PAN undergo significant deposition at velocities approaching 1 cm s$^{-1}$ over Amazonia and other forested areas, in consistency with field data (Sect. 3.5). It is even found that MVK (or MACR) deposits faster over forests than the considerably more soluble compounds CH$_3$OH and HCOOH. Caution is needed, however, in view of the large uncertainties, in particular for MVK and MACR, for which the model performance is highly variable against measurements (Table 5), with frequent occurrences of large under- and overestimations. Furthermore,

the existence of a large emission component to the measured fluxes of many OVOCs (methanol and formic acid in particular) might have led to an underestimation of the parameterized deposition velocities based on observed exchange velocities.



**Figure 7.** Global annually-averaged deposition velocity (cm s$^{-1}$) of (a) ozone, (b) PAN, (c) acetaldehyde, (d) acetone, (e) methanol, (f) formic acid, (g) MVK, (h) IEPOX, (i) HMHP.





**Table 6.** Global source and deposition fluxes of OVOCs in TgC/yr and TgN/yr, excluding the contribution of methane oxidation. The C deposition fraction is the ratio of the combined dry+wet deposition flux to the global sink (both expressed as carbon fluxes). The loss through aerosol uptake is also given. Notes: [a], including non-explicitly represented products; [b], global non-methane VOC emission flux.

| Compound | Source $TgC\ y^{-1}$ | Dry depos. $TgC\ y^{-1}$ | Wet depos. $TgC\ y^{-1}$ | Aerosol loss $TgC\ y^{-1}$ | C depos. fraction | Dry depos. $TgN\ y^{-1}$ | Wet depos. $TgN\ y^{-1}$ | Aerosol loss $TgN\ y^{-1}$ |
|---|---|---|---|---|---|---|---|---|
| $CH_3OH$ | 66.1 | 35.7 | 1.2 | | 0.56 | | | |
| $C_2H_5OH$ | 12.7 | 5.5 | 0.1 | | 0.44 | | | |
| ISOPBOH + ISOPDOH | 10.4 | 2.2 | 0.9 | | 0.30 | | | |
| HCOOH | 8.4 | 2.6 | 3.9 | | 0.77 | | | |
| $CH_3COOH$ | 31.4 | 7.0 | 9.5 | | 0.53 | | | |
| $CH_3OOH$ | 22.3 | 1.3 | 0.8 | | 0.09 | | | |
| HMHP | 3.4 | 1.2 | 0.6 | | 0.53 | | | |
| ISOPOOH | 114.4 | 10.7 | 5.5 | | 0.14 | | | |
| Peracids | 30.6 | 4.1 | 2.2 | | 0.20 | | | |
| HPALDs | 12.7 | 0.7 | 0.3 | | 0.08 | | | |
| Hydroperoxycarbonyls | 90.4 | 4.0 | 2.1 | | 0.07 | | | |
| Other hydroperoxides | 63.3 | 1.6 | 1.5 | | 0.05 | | | |
| IEPOX | 72.1 | 7.6 | 5.8 | 26.7 | 0.19 | | | |
| $CH_2O$ | 242.2 | 8.4 | 5.3 | | 0.06 | | | |
| $CH_3CHO$ | 93.8 | 16.1 | 0.03 | | 0.17 | | | |
| $CH_3COCH_3$ | 106.2 | 61.8 | 0.6 | | 0.59 | | | |
| MVK + MACR | 124.3 | 20.6 | 0.03 | | 0.17 | | | |
| HYAC | 37.9 | 5.4 | 2.7 | | 0.21 | | | |
| GLYALD | 34.3 | 3.1 | 2.6 | | 0.16 | | | |
| DHBO | 18.5 | 1.4 | 2.3 | | 0.20 | | | |
| GLY | 18.8 | 0.9 | 0.5 | 3.7 | 0.08 | | | |
| MGLY | 60.1 | 1.7 | 1.4 | | 0.05 | | | |
| BIACETOH | 16.2 | 0.6 | 1.2 | | 0.11 | | | |
| PAN-like compounds | 212.3 | 4.2 | 0.3 | | 0.02 | 2.39 | 0.17 | |
| ISOPN | 18.4 | 1.2 | 0.1 | 8.1 | 0.07 | 0.27 | 0.03 | 1.90 |
| MVKNO3 + MACNO3 | 3.6 | 0.3 | 0.3 | 0.05 | 0.19 | 0.10 | 0.10 | 0.02 |
| NISOPOOH | 3.3 | 0.3 | 0.05 | | 0.11 | 0.07 | 0.01 | |
| NC4CHO | 4.8 | 0.5 | 0.06 | | 0.11 | 0.11 | 0.01 | |
| NOA | 3.3 | 0.5 | 0.05 | | 0.18 | 0.21 | 0.02 | |



| Compound | Source TgC y$^{-1}$ | Dry depos. TgC y$^{-1}$ | Wet depos. TgC y$^{-1}$ | Aerosol loss TgC y$^{-1}$ | C depos. fraction | Dry depos. TgN y$^{-1}$ | Wet depos. TgN y$^{-1}$ | Aerosol loss TgN y$^{-1}$ |
|---|---|---|---|---|---|---|---|---|
| Monoterpenes[a] | 45.3 | $\sim$16 | $\sim$4 | $\sim$20 | $\sim$0.38 | 0.17 | 0.01 | 0.86 |
| Other organic nitrates | 8.9 | 1.0 | 0.2 | | 0.13 | 0.34 | 0.07 | |
| Other OVOCs | 82.0 | 5.1 | 2.0 | 13.1 | 0.09 | | | |
| ALL OVOCs | 855[b] | 235 | 58 | | 0.32 | 3.60 | 0.42 | 2.79 |

The calculated global dry and wet fluxes of OVOCs are summarized in Table 6. The relative importance of deposition is highly variable among the different species. The deposition fraction, i.e. the fraction of the total sink which is due to deposition, is highest for soluble, slowly-reacting compounds like methanol (56%), ethanol (44%), formic acid (77%), acetic acid (53%) and acetone (59%). For the carboxylic

acids, those fractions are similar to previous model estimates, e.g. 56–68% for $CH_3COOH$ (Paulot, 2011; Lin et al., 2014), and 73–82% for HCOOH (Paulot, 2011; Stavrakou et al., 2012; Millet et al., 2015). However, whereas we find wet deposition to dominate over dry deposition for those compounds (Table 6), dry deposition was found to be dominant by Paulot (2011) for both acids. Our dry deposition parameterization leads to longer global lifetimes with respect to dry deposition ($\sim$11–15 days) compared to e.g. Paulot (2011) ($\sim$6.5 days for both acids) and Khan et al. (2017) (6.4 day for $CH_3COOH$). Those longer lifetimes might help reducing the large reported model

underestimation of carboxylic acid abundance measurements (Paulot, 2011; Millet et al., 2015; Khan et al., 2017).

For most other OVOCs, dry deposition is dominant over wet scavenging. Regarding methanol, the high calculated deposition fraction appears consistent with study of Wells et al. (2014) using GEOS-Chem predicting a global deposition fraction of 61%, only slightly higher than our estimate. However, dry deposition over land is significantly stronger in our study (23% of the total sink, vs. 14.5% in Wells et al.), whereas our estimated contributions of wet deposition (2%) and oceanic dry deposition (27%) are comparatively weaker than in GEOS-

Chem (3 and 41%, respectively). The differences for oceanic dry deposition are difficult to interpret but could result from differences in the calculated methanol concentration field and/or in the parameterization of the gas-phase air-sea transfer velocity, which follows Johnson (2010) in the study of Wells et al.; as seen on Fig. 1, significant differences in $k_g$ are noted at Northern mid-latitudes, where methanol concentrations are highest. Over land, the parameterization of methanol deposition velocity in Wells et al. follows the general recommendation of Karl et al. (2010) for OVOCs, i.e. $f_0 = f_1 = 1$, resulting in weaker deposition than in our study (by $\sim$20% over tropical forests and by >60% over

mid-latitude areas).

Regarding ethanol, our estimated contribution of dry deposition to the global sink (43%) is about twice larger than in the modelling studies of Millet et al. (2010) (<23%) and Naik et al. (2010) (25%). For acetone as well, dry deposition to land is found to be a stronger sink (20% of the total) than in previous studies, e.g. only 8% of the global sink in the study of Fischer et al. (2012) which assumed a deposition velocity over land (0.1 cm s$^{-1}$) much lower than our model estimates (Fig. 7). The latter are acknowledged to be very uncertain given the scarcity

of field measurements (Table 5). The global acetone deposition flux over land is 21 TgC yr$^{-1}$, more than a factor of three higher than in the study of Safieddine et al. (2017).

The deposition fraction is unsurprisingly lower for more reactive compounds, but it remains significant, e.g. between 14 and 21% for major isoprene oxidation products such as MVK and MACR, ISOPOOH, IEPOX, HYAC, GLYALD, DHBO and $CH_3COOOH$. For acetaldehyde as well, dry deposition is found to account for $\sim$17% of its global sink, i.e. about an order of magnitude more than in the study of Millet et al.

(2010). Due to its high solubility and high deposition velocities over vegetation (Fig. 7), deposition represents more than half of the global sink of HMHP, thereby significantly affecting the estimated formic acid production through HMHP due to alkene ozonolysis (Müller et al.,



2018). For very reactive and photolabile OVOCs such as hydroperoxycarbonyls and conjugated carbonyls, the deposition fraction is low but non-negligible (5–10%).

Given the crudeness of the monoterpene oxidation mechanism, a rigorous determination of the associated deposition fraction is currently out of reach. The non-explicit products are assumed here to be lost through deposition and SOA formation in similar amounts, whereas the

formation of gas-phase products non-represented in the mechanism (mostly $CO_2$) accounts for the remainder.

Overall, the dry and wet deposition of gas-phase NMVOC oxidation products is found to account for 34% of the global NMVOC emission flux on a carbon basis (Table 5 and Fig. 8); dry deposition alone accounts for most (80%) of that deposition flux, or about 27% of NMVOC emissions. Our calculated contribution of dry deposition is higher than in the study of Safieddine et al. (2017) using GEOS-Chem ($\sim$20% of NMVOC emissions), but it is consistent with the lower limit estimate ($\sim$20%) of Karl et al. (2010) based on MOZART model calculations

(with $f_0 = f_1 = 1$ in Wesely's formulation); their estimated sink through OVOC wet deposition is however 2–3 times larger than in our calculation, which results in a larger overall impact of deposition in their model. An assessment of the wet deposition scheme for OVOCs is warranted, but is out of scope of the present study.

Different processes might increase or decrease the estimated fraction of NMVOC emissions lost to dry deposition. On one hand, the still unsufficiently documented re-emission of chemical compounds by vegetation following the uptake and chemical conversion of other

compounds might play an important role; examples include the conversion of ISOPOOH to MVK or MACR (Misztal et al., 2016), the degradation of MVK/MACR into e.g. MEK or isobutyraldehyde (Cappellin et al., 2017; Muramoto et al., 2015), and the suggested degradation of HMHP into formic acid (Nguyen et al., 2015), for which evidence is however still lacking. In all those cases, dry deposition does not lead to a permanent carbon loss from the atmosphere, only to a transformation.

On the other hand, simplifications in the oxidation mechanism might lead to an underestimation of total deposition fluxes: although the

mechanism is generally carbon-conserving (taking $CO_2$ formation into account even though it is not explicitly written), the simplification of complex mechanistic steps inevitably implies neglecting depositional losses of non-radical intermediates. Those intermediates being generally highly oxidized and very reactive, their deposition sink should generally be small, of the order of $\sim$10%.

Finally, additional OVOC loss is due to SOA formation and subsequent deposition. Being crudely represented in the model, it is too uncertain to report here, although the particulate deposition sink is expected to be of lesser importance than the deposition of gas-phase

organics (Knote et al., 2015).

As seen on Fig. 8(b), the deposition of OVOCs (nitrates and PANs) represents a significant sink of NOx, amounting to 15–30% of NOx emissions over tropical rainforests, whereas this fraction most often does not exceed 10% during summer over Western Europe, Eastern China and the Northeastern U.S. Over boreal forests, the distribution shown on Fig. 8(b) becomes unreliable due to the longer lifetimes of nitrates and especially pernitrates resulting in important transport effects; the average deposition fraction for Siberia (35-65° E, 54-66° N) in

summer is 17%, and is primarily due to the dry deposition of PAN-like compounds (10% of NOx emissions).

Over the Southeastern U.S. (defined as 80-94.5° W, 29.5-40° N) during August-September 2013, the net loss of NOx to organic nitrates through either deposition or aerosol hydrolysis is estimated to be 22% of the regional NOx emissions using the MAGRITTE regional simulation over the U.S. This result agrees very well with model calculations by Fisher et al. (2016), in spite of important differences in the treatment of organic nitrates, in particular regarding heterogeneous losses (Müller et al., 2018). This NOx sink is primarily due to aerosol

uptake (for two thirds), the remainder being due to deposition. The deposition of PAN-like compounds is an additional NOx sink amounting to $\sim$5% of the emission over this region, somewhat higher than the previous estimate for the Eastern U.S. (3.3%) by Mao et al. (2013). Interestingly, the model resolution has a small influence on the results: for example, the total NOx sink due to OVOC deposition amounts to 12.0% and 12.5% of NOx emissions in the regional simulation (0.5° resolution) and in the global model run (2°×2.5°), respectively.




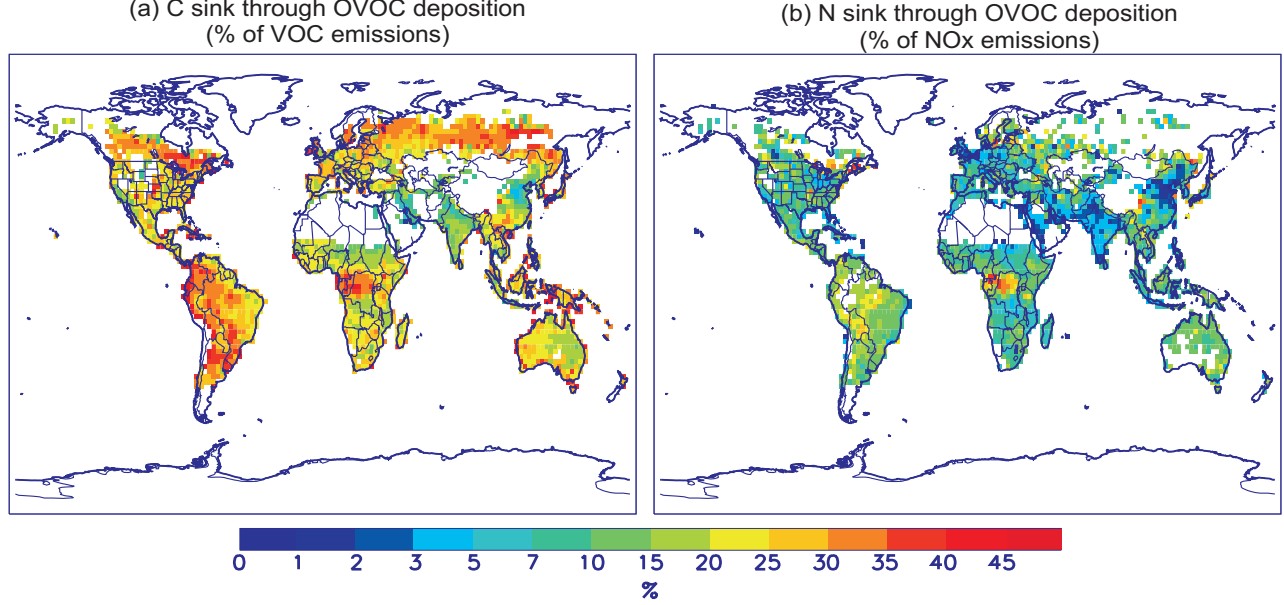

**Figure 8.** (a) Percentage ratio of carbon flux through OVOC deposition to the total annual NMVOC emission. (b) Percentage ratio of nitrogen flux through OVOC deposition sink to the total annual NOx emission.

At the global scale, we estimate that the deposition of organic nitrates and PANs removes 1.5 and 2.6 Tg(NOx-N) yr$^{-1}$ from the atmosphere, respectively, whereas the $RONO_2$ aerosol sink converts 2.8 Tg(NOx-N) yr$^{-1}$ into $HNO_3$ (Table 6). Therefore, out of the 50 Tg(NOx-N) yr$^{-1}$ emitted globally, it is estimated that about 8% and 5% are lost through OVOC deposition and hydrolysis, while the remainder is lost through $NO_2 + OH$ and the hydrolysis of $N_2O_5$ on aqueous aerosols. The relevance of PAN deposition to the budget of NOx is

frequently overlooked, because it represents only a tiny fraction (1.9%) of the global sink of PAN. Note that this fraction was slightly lower (1.2%) in the model study of Fischer et al. (2014) which adopted $f_0 = f_1 = 1$; PAN deposition was neglected entirely in Khan et al. (2017). Due to the very large PAN production, however, PAN deposition is estimated here to be a NOx sink of importance comparable to $RONO_2$ deposition.

## 5    Conclusions

We present a Wesely-type scheme for the calculation of dry deposition velocities of gaseous compounds in large-scale atmospheric models, with a particular emphasis on OVOCs. The scheme has four main components:

• aerodynamic resistance calculation (Eq. 4-12) based on a similarity-theory-based representation,

  • quasi-laminar boundary layer resistance (Eq. 17) calculation,

  • oceanic surface resistance calculation (Eq. 23-27) accounting for reactive uptake (Wesely, 1989) and solubility (Nightingale et al., 2000),

• canopy resistance calculation (Eq. 19) adapted from Wesely (1989) and Zhang et al. (2002),

Several essential subcomponents include

• species-dependent gaseous diffusivity calculation (Eq. 18) from molecular structure (Fuller et al., 1969),





• Henry's law constant calculation from molecular structure, using a newly-proposed estimation method (Sect. 2),

• stomatal resistance calculation (Eq. 28-31) adapted from MOHYCAN (Müller et al., 2008),

• cuticle and ground resistance calculation (Eq. 32-44).

The deposition scheme is evaluated against a large number of deposition velocity field measurements for OVOCs and for the model
compounds $O_3$ and $SO_2$ involved in the parameterization of the canopy resistances for OVOCs (Eq. 42-44). The evaluation accounts for the local LAI, vegetation type and sampling height, with ECMWF Era-Interim analyses providing the necessary meteorological input data.

Regarding $O_3$ and $SO_2$, the comparisons show essentially no bias, on average, for each major PFT (broadleaf decidous forest, coniferous forest, tropical forest, grass, crops). Furthermore, the model succeeds generally fairly well in reproducing the observed diurnal and seasonal cycles. This agreement is made possible by a significant adjustment of the parameters of the Jarvis-type module used for computing the
stomatal resistances (Eq. 29-31 and Table 3.2) in the canopy environment model MOHYCAN. Without this adjustment, the model would substantially underestimate the deposition velocities over temperate and tropical broadleaf forests. Nevertheless, the model does not reproduce the observed strong seasonal variation of $v_d^{O_3}$ at a rainforest site in Brazil, due to a substantial cold bias (3–5 K) in the ERA-Interim temperatures leading to an underestimated effect of drought stress on stomatal resistances. A more extensive evaluation of ERA-Interim daytime temperatures is crucially needed to determine whether such biases are typical or not over tropical forests and other ecosystems.

Regarding OVOCs, the model shows, on average, very little bias for every compound for which data is available (Table 5). The large scatter found in the model/data differences is to be expected, given the oversimplified representation of deposition in the model, the errors in meteorological input data, the representativity issue with measurement footprints often covering more than one landscape type, and the large uncertainties in observation-based deposition velocity estimates, in large part due to the presence of other processes influencing the fluxes, such as biogenic emissions and within-canopy chemical production or loss reactions.

The good model agreement with averaged observed OVOC deposition velocities is realized thanks to the adjustment of the model parameter $f_1$ relating the OVOC cuticular, mesophyll and ground resistances to the corresponding resistances for $SO_2$. A model simulation adopting $f_0 = f_1 = 1$ as recommended by Karl et al. (2010) greatly underestimates dry deposition for most OVOCs, by up to a factor of 4 for some species (see e.g. Fig. 6). As previously noted by Nguyen et al. (2015), the fastest-depositing compounds are generally very soluble in water, suggesting that the Henry's law constant (HLC) should be given a more prominent role than in the original resistance model of
Wesely (1989) and Zhang et al. (2002); this can be achieved e.g. by increasing $f_1$ or by modifying the resistance expressions to a similar effect (Nguyen et al., 2015). However, it is found impossible to match the observations for all OVOCs with a unique set of expressions for all species. Many species (e.g. PAN, acetone, acetaldehyde, MVK+MACR) are seen to deposit faster than expected based on their HLC, requiring higher $f_1$ values than for other, more soluble compounds. Caution is needed given the difficulties of observation-based deposition velocity estimates and the large diversity of observed behaviours between different sites. Still, the choice made here to adjust $f_1$ based on
measurements ensures the consistency of the model with reported field data, even though the mechanisms behind the deposition processes are poorly understood. Other choices of $f_0$ and $f_1$ parameter values could have realized a similar level of agreement with the data; or, the resistance parameterization based on model species ($O_3$ and $SO_2$) could be revised and replaced by a more mechanistic description, including separate representations for the different deposition pathways. Unfortunately, the deposition process is still poorly understood at the biochemical level, deposition measurements are too scarce for most OVOCs, and they are often poorly characterized. Refinements of the
current approach such as proposed in this work might be possible through a more thorough (although tedious and time-consuming) analysis of existing data, taking advantage of the relevant ancillary data including e.g. meteorological variables, friction velocity, roughness length, stomatal conductance, etc. These data are often measured but not always reported, and are generally not readily available in digital form. Looking ahead, it would be beneficial to the community to have access to the full measurement datasets after publication of field studies.





Implementation of the dry deposition scheme in the MAGRITTE chemistry-transport model provides an updated estimation of the contribution of dry (and also wet) deposition to the budget of OVOCs. Compared with previous studies, dry deposition appears to be a less efficient sink of the carboxylic acids, with longer global residence times (>11 days). For the other OVOCs, however, the calculated share of dry deposition to their global sink is generally larger than previously estimated, especially over land, with possibly important consequences

for top-down estimations of OVOC emissions based on atmospheric observations. In particular, dry deposition accounts for more than 50% of the global sink of methanol, acetone, hydroxymethylhydroperoxide, formic and acetic acid.

Overall, the dry deposition of OVOCs is calculated to remove from the atmosphere the equivalent of 27% of the global NMVOC emissions (on a carbon basis), excluding the contribution of SOA formation. Furthermore, deposition of organic nitrates and pernitrates is a significant sink of nitrogen oxides, estimated here at 8% of the global NOx source.

Uncertainties remain high, due to mechanistic simplifications, especially regarding monoterpene oxidation, and to the still incomplete understanding and crude representation of the dry deposition processes, including for example the conversion of deposited OVOCs to other species released into the atmosphere. Finally, the deposition velocities and Henry's law constants of highly oxygenated polyfunctional compounds remain poorly constrained by observations, whereas there is some evidence (Sect. 3.5.8-3.5.9) of large HLC overestimations by group contribution methods, likely due to H-bonding effects.

*Code and data availability.* The fortran code of the deposition scheme is available at http://tropo.aeronomie.be/index.php/models/magritte (doi:10.18758/71021042, last access: 15 December 2018). The compilation of dry deposition velocity measurements is made available in ascii format at the same webpage. Other relevant subroutines of the MAGRITTE model can be made available upon request. The MODIS leaf area index products are available from http://reverb.echo.nasa.gov/ (last access: 14 December 2018). The MAGRITTE model output is available upon request.

*Author contributions.* JFM designed and coded the deposition model, JFM and TS made the MAGRITTE model calculations, SC designed and evaluated the Henry's law constant prediction method, JFM and MB evaluated the model against deposition measurements, JFM and JP analyzed the results and drafted the manuscript.

*Competing interests.* The authors declare that they have no conflict of interest.

*Acknowledgements.* This research was supported by the Belgian Science Policy Office through the projects TROVA (2016–2018) within
the ESA/PRODEX programme, OCTAVE (2017–2021) within the BRAIN-be research programme, and BIOSOA within the SSD program (2006-2010).





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
