# Peer review of "S1 Evaluation of HLC estimation methods"

_Geoscientific Model Development, 2018_

## Referee Comment (RC1) · Anonymous Referee #1 · 21 Jan 2019

The study claims to have developed a new dry deposition scheme for calculating dry deposition velocities (Vd) of trace gases and implemented the scheme in MAGRITTE v1.0. Such a research activity should be encouraged considering the large uncertainties between existing schemes. The authors clearly have done a large amount of work. I have several major concerns listed below for the authors to consider in the revised version of the paper.

[Figure]

The motivation of the study needs to be better justified. Why is a new scheme needed while there are many existing ones, most of which have gone through extensive evaluation? How much improvements will this new scheme make compared to the existing ones? The uncertainties between some excising schemes have been demonstrated, e.g., in North America (Wu et al., 2018) and Europe (Flechard et al., 2011). These studies show that some schemes may provide reasonable Vd values in some circumstances. Will the new scheme have similar uncertainties? Will it provide better Vd values (in terms of mean bias, diurnal and seasonal patters, better correlations, etc., compared to available measurements) than existing schemes? A brief comparison of model performance can be made easily by simply examining published results.

The new scheme combines various formulas from many different existing schemes and ends up to be much more complex than most existing schemes. As is known, a more complex scheme does not necessarily provide better results than simple ones due to the more needed input parameters, which will likely bring in more uncertainties to the model output (Wu et al., 2018). Besides, a more complex scheme will have less attraction to the end-users for general applications.

The effective Henry's law constant (instead of the physical ones) should be used in all the cases (the deposition to leaf surface is controlled by the thin water film on leaf cuticles) (Zhang et al., 2002). This scheme seems to only consider such an effect in some cases (page 4 line 29).

This study is aware of the different conclusions from different measurement studies regarding the VOC Vd. Thus, adjusting some parameters to get a better agreement simply based on one-single site study may not proved reasonable results elsewhere (an approach frequently used in this study). For example, Karl et al. [2010] suggested that Vd of VOCs calculated using existing schemes is about a factor of 2 lower than those based on canopy-level concentration gradient measurements over six sites covering forests and shrublands. The relative magnitudes between Vd(VOCs) and Vd(O3) in Karl et al. [2010] are actually similar to those in Zhang et al. (2003) in that Vd(VOCs) is

slightly small than Vd(O3). While the typical daytime Vd(O3) over vegetated canopies is around 1 cm s-1 in literature from numerous studies (Silva and Heald, 2018), the value in Karl et al. [2010] is much higher (e.g., up to 2.4 cm s-1 at canopy top). Thus, the very high Vd(VOCs) presented in Karl et al. [2010] was likely caused by special chemical conditions, besides uncertainties in measurements and methodology used in estimating Vd, and may not be generalized to other canopies or even to the same type of canopy but in different regions. In addition, uncertainties in field measured fluxes could be larger than theoretically constrained values if the measurement method and instrument detection limit cannot satisfy the flux measurement requirements. For example, different measurement method provide quire different flux at the same site (Wu et al., 2015). By saying this, I do not recommend constructing a Vd scheme producing such high Vd values as shown in Figure 6 for those species before mere field flux data become available to support these high Vd values.

References mentioned above:

Flechard C.R., Nemitz E., Smith R.I., Fowler D., Vermeulen A.T., Bleeker A., Erisman J.W., Simpson D., Zhang L., Tang Y.S., and Sutton M.A., 2011. Dry deposition of reactive nitrogen to European ecosystems: a comparison of inferential models across the NitroEurope network. Atmospheric Chemistry and Physics, 11, 2703-2728.

Karl, T., P. Harley, L. Emmons, B. Thornton, A. Guenther, C. Basu, A. Turnipseed, and K. Jardine (2010), Efficient atmospheric cleansing of oxidized organic trace gases by vegetation, Science, 330(6005), 816-819.

Silva, S.J. and C.L. Heald (2018), Investigating Dry Deposition of Ozone to Vegetation. Journal of Geophysical Research: Atmospheres, 123, 559-573.

Wu, Z., D. B. Schwede, R. Vet, J. T. Walker, M. Shaw, R. Staebler, and L. Zhang (2018), Evaluation and intercomparison of five North American dry deposition algorithms at a mixed forest site, Journal of Advances in Modeling Earth Systems, 10(7), 1571-1586.

Wu Z., Zhang L., Wang X., and Munger J.W., 2015. A modified micrometeorological gradient method for estimating O3 dry deposition over a forest canopy. Atmospheric Chemistry and Physics, 15, 7487-7496.

Zhang, L., J. Brook, and R. Vet (2003), A revised parameterization for gaseous dry deposition in air-quality models, Atmos. Chem. Phys., 3(6), 2067-2082.

Zhang, L., M. D. Moran, P. A. Makar, J. R. Brook, and S. Gong (2002), Modelling gaseous dry deposition in AURAMS: a unified regional air-quality modelling system, Atmos. Environ., 36(3), 537-560.

---

## Referee Comment (RC2) · Anonymous Referee #2 · 28 Jan 2019

Review of Müller et al. General comments

Müller and colleagues report the development of a new parameterization for dry deposition of many different compounds. Much of the paper is dedicated to model evaluation of simulated deposition velocities — the authors pull available observations for different compounds from peer-reviewed literature, and examine mean diel cycles at different locations, sometimes for different seasons or months. They report global

sources and sinks for different OVOCs, and the C loss through OVOC deposition vs. NMVOC emission and N loss through OVOC deposition vs. NOx emission. I'm not sure how meaningful the latter analysis is. The authors find pretty substantially different budget terms for OVOCs than previously reported in the literature. In terms of the global budget analysis, it is not clear why other compounds (ozone, SO2, etc) are not included in the latter analysis.

Their new parameterization for dry deposition is a bit strange. For stomatal conductance, but not for nonstomatal cuticular deposition, which also varies with height in the canopy due to LAI, there is some form of a multilayer canopy model. However, the multilayer canopy model does not consider variations in in-canopy turbulence - suggesting that depositing compounds get to plant stomata at all levels of the canopy with complete ease. The authors suggest in the conclusion that the model agreement with observations very much so stems from this stomatal conductance parameterization, so it seems like this component needs to be evaluated more carefully. The authors implement mostly the Zhang et al. 2002, 2003 equations for nonstomatal deposition. These equations are relatively widely used, but are based on a regression with relatively few short-term datasets, all from the eastern United States. I would like to see this at least mentioned in the paper.

Müller and colleagues do not seem to appreciate that depositional processes are highly uncertain, and that observational evidence consistently shows that stomatal deposition is not the only driver of variations in deposition velocities. The model evaluation section is replete with entirely hand-wave-y explanations for variations in the observations and model biases, and I find the authors to be a bit overconfident in terms of how trustworthy their simulated deposition velocities are. I have a lot of comments below for the model evaluation, but they do taper off in that section, not because I feel like the section is good enough, but because my comments are similar to what I've already said and I do not feel it is my duty to "fix" this paper. I think this paper needs extensive work before it is suitable for publication.

Line-by-line comments Page 1 Line 7: phrasing respect to LAI is confusing; what accounts for the LAI? The model simulates the LAI? Line 13: there are plenty of papers suggesting important roles for nonstomatal deposition processes driving variations and the magnitude of the ozone deposition velocity Line 20-1: Is the caveat here that the authors tune the model to the data? If so, please just cut this line. otherwise, please give an example of what the caveat(s) are.

Page 2 Line 5: Wesely (1989) doesn't actually show this result Line 15: approach "from" Wesely (1989) might be clearer; suggest adding (hereafter, referred to as "Wesely") or something like it Line 20: "relatively" abundant Line 22: "with" the exception Line 23: Can the authors repeat the view that they are talking about here for the reader? Line 32: model adaption of what?

Page 3 Line 12: role of dry deposition of which species? Also "on" instead of "in" for "in the budget..."? Line 16: specify the parameterizations in MAGRITTE? Say the authors will abbreviate Henry's law constant as HLC? Line 17: why do the authors refer to the scheme as a wet scavenging scheme if the scheme has both wet scavenging and washout by precipitation? Line 25: "chemical"

Page 4 Line 8: Not sure what "within its scope" refers to Line 10: organic part of what? What is "it"? Line 12: Not sure what "has the widest scope of all methods" means Line 14: where in the supplement? Line 17: by degrades the results, do the authors means the model evaluation against the observations? Line 20: "due to the limited scope of their basis sets of empirical data" is confusing

Page 8 Line 2: A bit awkward to say "Following eg Wesely" Line 5: Rb isn't only controlled by diffusivity Line 10: Not sure I would say they are well represented Line 27: Why is the reduced reference height given the same acronym as the reference height? Line 27: I don't think this is right; I think the model should be compared with the sampling height? What is the assumption here? Line 28-9: give some indication why it's important then? Line 29-31: I'm confused by this statement. What does "it

takes generally much lower values" mean...?

Page 9 Line 26: "taken to be equal to" Line 30: Are year-specific LAI values used?

Page 10 Table 2: Include names of acronyms in title of table Line 4: citation for "although ... water" Line 14: experiment-based estimates of what?

Page 11 Line 1-2: perhaps define Rg as ground uptake and include a little more discussion of why over oceans Rc=Rg. Depending on the parameterization of Rg over land (if it depends on soil moisture or organic content), it might be confusing to say Rc=Rg over oceans. Line 11: can the authors expand a bit on a gas-phase vs. liquid phase transfer velocity? Are some of the compounds being exchanged in the liquid phase? Line 14-15: this is a bit confusing - there seem to be different thoughts going on here - perhaps separating whats been done before v.s. what the authors are doing in their model would help Line 15: "those"-> "oceanic" Line 17-21: I'm not sure the point of this analysis. Is Johnson (2010) regarded as the truth? If so, why? Is this essentially a comparison of models for 1/(Ra+Rb)? If the authors do indeed retain this analysis it would be helpful if the masked out the land in Figure 1a for ease of comparison between a and b Line 22: Can the authors explicitly say where Rc falls in this parameterization for the ocean? I'm not seeing it in equations 20-22. Line 26: "is minor except for highly reactive but poorly soluble compounds" Line 27: What is the physical meaning of the f0?

Page 12 Line 4: use of "latter" is confusing; which parameter are the authors referring to? Why is it's on the order of relevant here? Line 14: description of mesophyllic resistance is too vague; can the authors be more specific? Line 17: back and forth of referring to as Rm vs. mesophyll resistance; why does a high deposition velocity for OVOCs necessarily imply low Rm? There could be high nonstomatal deposition

Page 13 Line 5: it's flawed to have a multilayer canopy model for just stomatal conductance, and not the other nonstomatal deposition pathways; additionally this equation for stomatal conductance doesn't take into account canopy turbulence, and why is f1=1

for shrubs and 1.25 for other PFTs? I assume this parameter is trying to account for whether one side of the leaves, or both sides of the leaves have stomata on them. These numbers seem to be chosen at random; where are the authors getting the vertical distribution of LAI in the canopy? Line 12: "are evaluated"-> "are calculated" Line 15: So, a_s, b_s, c_s, and d_s are tuning parameters? Where are they from? How are they inferred? Line 20: why is val Martin et al. 2014 cited here? The authors should cite the original papers that describe the observations that val Martin et al. use to evaluate CESM Line 23-4: citation needed for strong dependence of deposition velocity on water vapor deficit here (do the authors mean vapor pressure deficit?); the back and forth of referring to future sections is frustrated, and is making me think that the authors are tuning the parameterization to the observations they collect from the literature, which I don't think is the case. This could perhaps be alleviated by briefly mentioning what is discussed in future sections Line 24: Zhang et al. 2006 is definitely not a sufficient citation for the dominance of stomatal uptake for ozone during daytime Line 26: Again, why is Val Martin et al. 2014 cited here? They did not take any measurements; Also, I don't think it's really fair to say it's typically on the order of 100 s/m on the basis on just a few observational results Lines 29-32: I appreciate that the authors take the time to discuss this but it also merits mention that these parameters and equations are typically only based on a few observations for each PFT

Page 14 Line 1: there is actually a fair amount of evidence suggesting that ozone deposition is not enhanced on wet leaf cuticles — see Massman (2004) Line 2: important to acknowledge that Zhang parameterization is based on regression of only a couple very short-term monitoring sites, all in the eastern US, equations don't necessarily represent processes Line 23-4: there is stronger evidence that R_g_O3 is high for moist soil and low for wet soil than for R_cut_O3 low for wet and high for dry — see Massman (2004) Line 26: "the resistance" —> Rg_SO2 Line 28: does soil pH vary temporally at all? Is the soil pH the same for all land cover/land use types within a certain grid cell? Line 30: where is this equation from?

Page 15 Line 1: where is this contribution from? Line 10: Wu et al. (2018) suggests this parameterization for f_snow is problematic at a temperate mixed forest Line 12: by "for any chemical compound" do the authors mean OVOCs, or all?

Page 16 Line 8: so the authors do opposite to what Karl et al. (2010) recommends. . . why? Line 12-4: will the authors please better explain what they are doing? It seems like they are calculating Ra based on z_l Line 15: albeit short-term campaigns for the most part Line 22: the authors are likely assuming very low Ra and Rac already — I doubt the underestimate is due to this; the Wu et al. 2011 hypothesis is not well supported Line 25-6: will the authors integrate these sentences into the discussion better? Are the authors talking about their model, or a model used in Rummel et al.? Line 26-7: are the authors actually doing any process-level investigation here? How can they attribute the overestimate to an underestimation of Ra? It might be helpful if the first two sentences of this paragraph are combined into one sentence. The rest of the paragraph is confusing and seemingly provides evidence against the authors hypothesis at the beginning of the paragraph. Line 34: the observed deposition velocities? Or the modeled? Line 36: Doesn't seem to me from this Fares paper that stomatal dominates the total. It is the highest contribution, but contributions from cuticular and soil are not far behind and the sum of them outweighs the stomatal contribution.

Page 17 Figure 2a - Clifton et al. 2017 shows all years at Harvard Forest. There is a lot of interannual variability, and Wu et al. 2011 only show one year. I would suggesting using the Clifton et al. 2017 results rather than the Wu et al. 2011 results Figure 2f - why is the gradient method even shown if there are EC observations? Line 1- this seems highly speculative

Page 18 Line 1: suggest period after "50%)" and new sentence starting with "In the model, this is due to" Line 5: cut "in optimal (unstressed) conditions"? That high deposition velocity could be due to a lot of things, not only stomatal conductance Line 5-8: suggest rearranging this sentence so it is not as leading in suggesting that sesquiterpenes dominate the ozone flux. Also how do the authors derive the 1 cm/s from the

reported fluxes? Line 9-11: not sure what the authors are getting at here Line 11: stretch to say "clearly responsible" - the authors have done no attribution work.

Page 19 Line 3: also a stretch to say what largely drives the seasonality; the authors are comparing two very short term time periods with data Line 10: can the authors give an estimate of the underestimated VPD?

Page 20 Lines 1-2: friction velocity form the model, or the obs? Line 3-4: best not to speculate here. Line 5-14: again best to re-frame this and not be so leading w.r.t. stomatal deposition driving differences Line 6 - why is this especially true for crops? Why are the parameters for shrub assumed to be the same as broadleaf deciduous trees? Line 10: why is Zhang cited here? Line 17: what are the critical resistances? Line 18: variable between species? Line 27: "conductance" —> "dry deposition" Line 27-8: clarify "in order to rationalize their similar deposition velocities and similarity with ozone"

Page 21 Line 1-2: citation? Line 4: citation? Line 7-8: clarify obs or modeled deposition velocities Line 9: Model of Wesely or Zhang? Which one is it? Are the authors examining both, or a hybrid approach? Line 11-12: what about nonstomatal deposition? "clearly indicate" is way too strong Line 20: Do the authors compare to the same datasets as Paulot et al. 2018? Do Paulot et al. evaluate the model in a similar fashion?

Table 4 Probably not a good idea to evaluate with branch enclosure measurements - how do the authors scale up? Generally, "see text" is not very helpful., Where in the text do the authors talk about this? (same thing for Table 5 - scaling up branch enclosure measurements just assuming Ra and Rb is probably not a good idea).

Table 5 What does corrected for non deposition fluxes mean? Where in the text do the authors discuss this? "see text" is not very helpful.

Figure 8: Does it make sense to look at these ratios? Are these ratios really telling how

much of what was emitted was deposited through OVOC emission?

Page 25 Line 1: I don't think the observations clearly suggest this. There is only one paper (Turnipseed et al.) that suggests this. I don't think the authors' model evaluation here is enough to say that the observations clearly support a role for liquid water ON leaves and needles. Line 2: what exactly is the presence of the double bond doing? Where is the double bond? In MPAN? Or in the things in the liquid water that MPAN is reacting with? Line 3: Can the authors draw this out a little more - what do they mean caution is warranted?

Page 26 Lines 1-13: it's pretty unclear what the authors are getting at here. I urge them to make this more clear Line 15: How do the authors figure out that it's due to an underestimate of stomatal conductance? Are these values the numbers for stomatal conductance for PAN, or deposition velocity for PAN? Line 16: I thought the authors were saying before that the model of Paulot et al. 2018 did worse. Line 20: can the authors elaborate on what exactly was involved in the "fumigation experiments" Line 21: so what do these results suggest? Line 33-5: the model values for what? Should this sentence be at the beginning of this paragraph?

Page 27 Line 5: "but by how much remains uncertain" Line 8: why not ground deposition? Line 13-15: Again this assessment seems too confidently expressed Page 32 Line 11: "inherit from"—> "are based on"; are these citations for the IMAGES model, or are they citations for the adaption of this model? Line 17-20: better articulate what the difference between these simulations actually represents? When is METHONLY actually used in the following sections?

Figure 7 - for which year?

Table 6 - what about gas-phase? chemical loss

Page 35 Line 1: What does "Relative importance of deposition" mean? The fraction of dry deposition/total loss? Please refer to it as dry deposition, unless the authors

are referring to both wet and dry deposition Line 10: "previously reported by models" Line 13: give number after "than our estimate" Line 17: "significant differences in kg between our parameterization and that of Johnson (2010)"

Page 37 Line 10: why is this model a "Wesely type" scheme? Line 10- Page 38, Line 3 - this list isn't very helpful for the reader

Page 38 Generally I think the authors are a bit too confident about their model evaluation Line 5-6: How does the model evaluation account for local LAI, vegetation type and sampling height? Please be more specific. It's likely that the re-analysis met fields are quite different than the flux-tower met data